# PromptRL: Prompt Matters in RL for Flow-Based Image Generation

Fu-Yun Wang [1 2 *]   Han Zhang [3]   Michaël Gharbi [2]   Hongsheng Li [1]   Taesung Park [2]

## Abstract

Flow matching models (FMs) have revolutionized text-to-image (T2I) generation, with reinforcement learning (RL) serving as a critical post-training strategy. We show that current RL pipelines for FMs suffer from two underappreciated limitations: sample inefficiency from collapsed generation diversity, and prompt overfitting, where models memorize training phrasings and collapse on semantically equivalent paraphrases. We present **PromptRL** (**P**rompt **M**atters in **RL** for Flow-Based Image Generation), a framework that incorporates language models (LMs) as trainable prompt refinement agents directly within the flow-based RL loop, yielding both a learned prompt rewriter and a synergistic training regime that reshapes optimization dynamics. PromptRL attains state-of-the-art scores of 0.97 on GenEval, 0.98 on OCR accuracy, and 24.05 on PickScore, and on instructional editing it improves FLUX.1-Kontext from 1.19 to 1.43 EditReward with only 0.06 M rollouts, surpassing Gemini 2.5 Flash Image (Nano Banana, 1.37) and matching ReasonEdit-Think (1.44), which relied on fine-grained data annotations and a multi-stage pipeline. Our experiments show that PromptRL achieves higher performance ceilings with approximately 50% fewer rollouts than naive flow-only RL. Our code is available at https://github.com/G-U-N/UniRL.

## 1. Introduction

The advent of flow matching models (FMs) (Liu et al., 2023; Lipman et al., 2023; Labs, 2024; Song et al., 2021; Wang et al., 2025b; 2024a) has transformed text-to-image (T2I) generation, enabling photorealistic synthesis from natural language descriptions (Mumuni et al., 2024). To align these models with human preferences and specific reward objectives, reinforcement learning (RL) (Sutton & Barto, 2018; Fan et al., 2023; Black et al., 2024) has become the standard post-training mechanism, refining model behavior beyond the scope of supervised pretraining. Despite these advances, applying RL to FMs remains prohibitively sample-inefficient.

Our investigation reveals two underappreciated yet critical failure modes in current flow-based RL pipelines. (*i*) First, we observe a counterintuitive *exploration paradox*: as T2I models improve at following prompts precisely, they simultaneously lose generative diversity under identical prompts. This increased prompt adherence constrains the behavioral variation necessary for effective RL exploration, causing optimization to stagnate in narrow modes of the generation space. (*ii*) Second, we identify severe *prompt overfitting*, where models learn to exploit superficial linguistic patterns in training prompts rather than developing genuine visual understanding. This overfitting manifests as dramatic performance collapse when models encounter semantically equivalent prompts phrased with different syntax at test time. This makes prompt enhancement (PE) (Hao et al., 2023; Rosenman et al., 2024; Mañas et al., 2024; Mo et al., 2024), a crucial technique for improving generation quality, ineffective or even counterproductive for RL-finetuned FMs. We provide detailed empirical evidence for both phenomena in Section 3.

These limitations expose a fundamental design oversight in existing approaches: treating prompts as fixed inputs rather than malleable components of the optimization process. Naive augmentation techniques such as random synonym substitution or rule-based paraphrasing prove inadequate, often failing to generate semantically coherent variations at scale. In this paper, we explore the hypothesis that large language models (LMs) (Cao et al., 2024), when trained as adaptive co-learners via joint RL, can generate semantically grounded prompt variations that enhance exploration efficiency and serve as a co-trained PE module in practical deployment.

We introduce **PromptRL** (**P**rompt **M**atters in **RL**), a framework that integrates LMs as adaptive co-learners within

---

[*]Work done during an internship at Reve.   [1]The Chinese University of Hong Kong, Hong Kong [2]Reve, USA [3]Meta Superintelligence Labs, USA. Correspondence to: Han Zhang <hanzhang.ai@gmail.com>, Hongsheng Li <hsli@ee.cuhk.edu.hk>.

*Proceedings of the 43rd International Conference on Machine Learning*, Seoul, South Korea. PMLR 306, 2026. Copyright 2026 by the author(s).

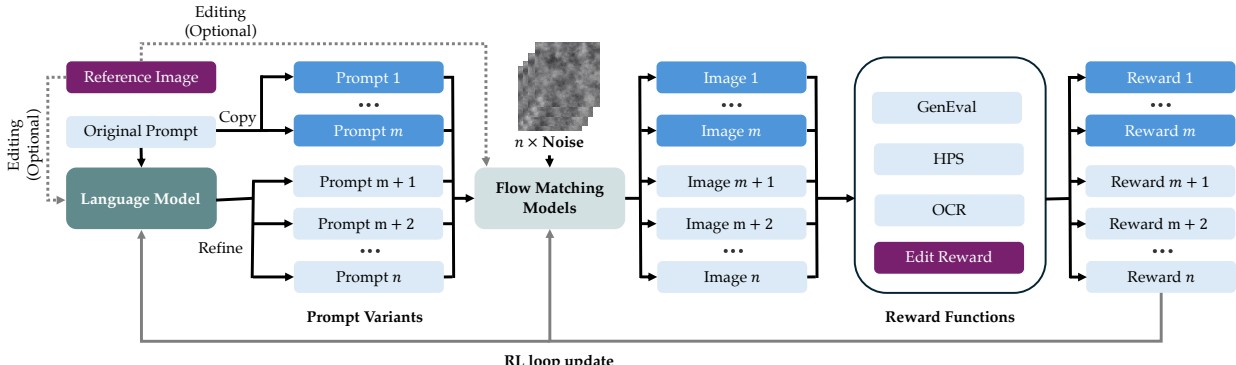

*Figure 1.* **Overview of the PromptRL framework.** PromptRL jointly trains a language model and a flow-matching image generator within a unified RL loop. Given an original prompt (and optionally a reference image), the LM produces semantically grounded prompt variants that expand the exploration space beyond fixed-prompt training. These prompts are paired with independent noise samples and passed to the flow-matching model to generate diverse images. A mixture of reward functions evaluates each image and guides the evolution of the LM (for improved prompt rewriting) and the FM (for improved visual generation).

flow-based RL training loops, as illustrated in Fig. 1. Rather than employing LMs as static preprocessors, we train them to generate prompt variations that simultaneously preserve semantic intent and maximize downstream image generation rewards. This creates a mutually beneficial training dynamic: diverse LM-generated prompts expand the exploration space for FMs, accelerating policy improvement, while reward signals from flow model outputs guide LMs toward discovering linguistically varied yet contextually appropriate reformulations.

Our experimental results demonstrate that PromptRL achieves state-of-the-art performance across multiple benchmarks, obtaining scores of 0.97 on GenEval (Ghosh et al., 2023), 0.98 on OCR accuracy (Cui et al., 2025), and 24.05 on PickScore (Kirstain et al., 2023). Furthermore, we validate the effectiveness of our RL approach on large-scale image editing models, improving EditReward (Wu et al., 2026a) of FLUX.1-Kontext (Black Forest Labs et al., 2025) from 1.19 to 1.43 with only 0.06 million rollouts. Extensive experiments show that PromptRL consistently achieves higher performance ceilings while requiring approximately 50% fewer rollouts compared to existing methods, all while maintaining robust generalization to diverse prompt formulations. These findings establish language-vision co-optimization as a foundational principle for efficient and robust preference learning in generative models.

**Conflict of Interest Disclosure.** The authors declare no financial conflicts of interest related to this work. All baselines and reward models evaluated in this paper (*e.g.*, FLUX.1-dev, FLUX.1-Kontext, Stable Diffusion variants, Gemini 2.5 Flash Image, ReasonEdit, PickScore, GenEval, EditReward) are external systems not developed by, or affiliated with, the authors' employing institutions. No internal commercial product of any author affiliation is used as a

benchmarked system, training reward, or evaluation oracle.

## 2. Related Works

**RL for image generation.** Reinforcement learning for flow-based image generation has evolved through several paradigms. Early differentiable reward methods (*e.g.*, DRaFT (Clark et al., 2024), AlignProp (Prabhudesai et al., 2023), ReFL (Xu et al., 2023)) backpropagate gradients from pre-trained reward models, offering simplicity but proneness to reward hacking such as oversaturation. RL-based approaches (*e.g.*, DDPO (Black et al., 2024), DPOK (Fan et al., 2023)) treat denoising as an MDP and apply PPO or variants for alignment, with scaled versions showing promise (Zhang et al., 2024). Direct preference optimization advanced with Diffusion-DPO (Wallace et al., 2024), later extended by D3PO (Yang et al., 2024), SPO (Liang et al., 2025), and DiffusionNPO (Wang et al., 2025a), the last using negative preferences to suppress undesirable modes. Flow-GRPO (Liu et al., 2025a) and DanceGRPO (Xue et al., 2025) extend DDPM-based RL to flow matching by converting flow ODEs to SDEs, while DiffusionNFT (Zheng et al., 2026) performs efficient online RL on the forward noising process by contrasting positive/negative samples without reverse gradients. Mix-GRPO (Li et al., 2025) improves GRPO efficiency via hybrid ODE–SDE sampling. Joint RL on unified LM and diffusion experts has also been explored (Wang et al., 2025c).

**PE for image generation.** Prompt enhancement (PE) has become essential for improving T2I generation quality. Recent LM-based methods include Promptist (Hao et al., 2023) (SFT+RL for aesthetics), NeuroPrompts (Rosenman et al., 2024) (constrained decoding), OPT2I (Mañas et al., 2024) (iterative LM refinement), RePrompt (Wu et al., 2026b) (CoT reasoning with reward-guided training), and PAE (Mo

| Prompt | (a) Stable Diffusion v1-5 | (b) FLUX.1-dev | (c) FLUX.1-dev + LM refinement |
|---|---|---|---|

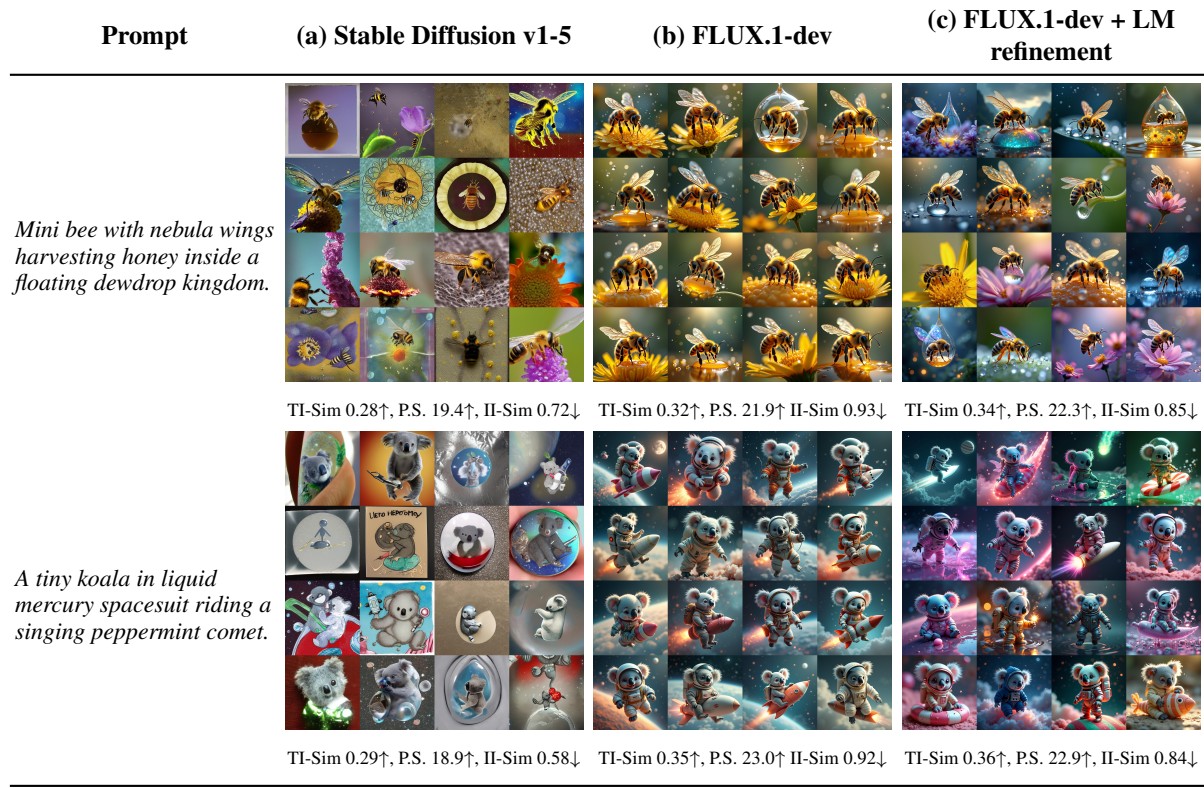

*Mini bee with nebula wings harvesting honey inside a floating dewdrop kingdom.*

TI-Sim 0.28↑, P.S. 19.4↑, II-Sim 0.72↓    TI-Sim 0.32↑, P.S. 21.9↑ II-Sim 0.93↓    TI-Sim 0.34↑, P.S. 22.3↑, II-Sim 0.85↓

*A tiny koala in liquid mercury spacesuit riding a singing peppermint comet.*

TI-Sim 0.29↑, P.S. 18.9↑, II-Sim 0.58↓    TI-Sim 0.35↑, P.S. 23.0↑ II-Sim 0.92↓    TI-Sim 0.36↑, P.S. 22.9↑, II-Sim 0.84↓

*Figure 2.* **The quality-diversity dilemma in flow-based T2I models and its mitigation through prompt refinement.** As models advance from Stable Diffusion v1-5 (a) to FLUX.1-dev (b), they achieve higher text-image alignment (TI-Sim) and aesthetic quality (P.S.) but suffer from dramatically reduced output diversity (II-Sim), creating an exploration bottleneck for RL optimization. LM-based prompt refinement (c) partially restores diversity while maintaining quality, demonstrating that linguistic variations can expand the exploration space. All images in each row share identical random seeds to isolate the effect of prompt conditioning.

et al., 2024) (joint word-weight and timestep optimization). These methods are limited to prompt-level RL with modest gains—for instance, RePrompt only improves FLUX.1-dev's GenEval from 0.66 to 0.76—and do not co-optimize prompts with the underlying FM. For instructional editing, MGIE (Fu et al., 2024), SmartEdit (Huang et al., 2024), and Emu Edit (Sheynin et al., 2024) similarly leverage LMs to expand vague user instructions into expressive editing guidance.

## 3. Understanding Flow RL Inefficiencies

### 3.1. Generation Dilemma: Quality *vs*. Diversity

We observe a fundamental tension between generation quality and output diversity in modern T2I FMs. As models advance in their capacity to precisely follow textual prompts, they simultaneously sacrifice the generative variability essential for effective RL exploration. To quantify this phenomenon, we measure text-image similarity (TI-Sim) using CLIP ViT-g-14 (Radford et al., 2021) for prompt alignment, PickScore (P.S.) for aesthetic quality, and image-image similarity (II-Sim) via CLIP ViT-g-14 for output diversity under identical prompts. Fig. 2 demonstrates this quality-diversity

trade-off: Stable Diffusion v1-5 (column a) produces generations with moderate TI-Sim and P.S., but maintains substantial diversity (II-Sim of 0.58–0.72). In contrast, FLUX.1-dev (column b) achieves notably higher aesthetic scores, yet generates outputs with considerably reduced diversity (II-Sim of 0.92–0.93). This pattern holds consistently across semantically diverse prompts, from fantastical creatures to surreal compositional scenes. Notably, LM-based prompt refinement (column c) partially mitigates this collapse, reducing II-Sim to 0.84–0.85 while preserving quality metrics.

This quality-diversity dilemma directly undermines RL optimization in flow-based models. As generation policies become increasingly deterministic, rollout trajectories collapse into narrow modes of the output space, causing reward signals to degenerate: when all samples cluster around similar high-quality outputs, advantage estimators lose the comparative information necessary for policy improvement. This exploration bottleneck is further compounded by severe prompt overfitting, wherein models exploit superficial lexical patterns rather than semantic understanding, which we investigate in the following subsection.

*Table 1.* Performance comparison across different models and evaluation metrics. All metrics are tested with 20 Euler steps with resolution 1024. PE denotes prompt enhancement with Qwen2.5-VL. Green and red numbers indicate performance changes after applying PE. P.S. and U.R. denote PickScore and UnifiedReward, respectively.

| Model | w/ PE | GenEval | OCR | P.S. | HPS | U.R. |
|---|---|---|---|---|---|---|
| SD3 | ✗ | 0.58 | 0.48 | 22.30 | 26.95 | 2.982 |
| SD3 | ✓ | 0.63 (+0.05) | 0.53 (+0.05) | 22.34 (+0.04) | 27.67 (+0.72) | 3.140 (+0.158) |
| DiffusionNFT | ✗ | 0.88 | 0.89 | 23.63 | 31.78 | 3.392 |
| DiffusionNFT | ✓ | 0.77 (-0.11) | 0.86 (-0.03) | 23.21 (-0.42) | 30.66 (-1.12) | 3.268 (-0.124) |
| FlowGRPO | ✗ | 0.92 | 0.89 | 23.33 | — | — |
| FlowGRPO | ✓ | 0.81 (-0.11) | 0.86 (-0.03) | 23.13 (-0.20) | — | — |

## 3.2. Prompt Linguistic Hacking

Beyond the quality-diversity dilemma, we identify a second critical failure mode: *prompt linguistic hacking*, where RL-trained models exploit superficial lexical patterns rather than developing robust semantic understanding. We evaluate this by testing models on both original prompts and semantically-preserved paraphrases generated by Qwen2.5-VL (*i.e.*, PE) (Bai et al., 2025).

As shown in Table 1, the pretrained SD3 (Esser et al., 2024) demonstrates linguistic robustness, with consistent or improved performance under paraphrasing across all metrics. However, flow-only RL models exhibit severe prompt overfitting. DiffusionNFT achieves strong performance on original prompts but suffers catastrophic degradation under paraphrasing. Similarly, FlowGRPO trained on GenEval drops from 0.92 to 0.81 when prompts are paraphrased. This indicates that *learned policies memorize superficial linguistic features* rather than understanding underlying visual concepts. This overfitting occurs at the *prompt-conditioning* level and cannot be resolved through standard regularization. More critically, *PE techniques that benefit pretrained FMs become ineffective or even harmful after flow-only RL*, as fine-tuned models overfit to specific prompt distributions. This observation motivates our joint LM-FM optimization approach: rather than applying PE as a fixed preprocessing step, we co-evolve the prompt enhancer and generator in a symbiotic manner.

## 4. PromptRL

### 4.1. Incorporating LMs as Dynamic Prompt Refiners

To address the dual challenges of exploration collapse and prompt overfitting identified in Section 3, we propose incorporating LMs as adaptive prompt refiners within the RL training loop. Unlike static augmentation techniques that rely on rule-based transformations or synonym substitution, our approach leverages the semantic understanding and compositional flexibility of pretrained LMs (Zhao et al., 2025) to generate contextually grounded prompt variations.

Formally, given an original prompt $p_0$ from the training distribution, we deploy an LM $\pi_{\text{LM}}(\cdot|p_0)$ to generate a set of refined prompts $\{p_1, p_2, \ldots, p_k\}$ that preserve the core semantic intent while introducing linguistic diversity. Each refined prompt $p_i$ is then paired with an independent noise sample $\epsilon_i \sim \mathcal{N}(0, I)$ and fed into the flow-matching model $\pi_{\text{FM}}(\cdot|p_i, \epsilon_i)$ to produce diverse image samples $\{\mathbf{x}_1, \mathbf{x}_2, \ldots, \mathbf{x}_k\}$. This architecture creates a hierarchical exploration mechanism: the LM explores the linguistic manifold of semantically equivalent descriptions, while the FM explores the visual manifold conditioned on each prompt variant.

Critically, we introduce a *prompt retention mechanism* during training: for each batch of $n$ total samples, we retain $m < n$ samples that use the original prompt $p_0$ without LM refinement, while the remaining $n-m$ samples undergo LM-based augmentation. This design serves two complementary purposes. First, unmodified prompts provide a *strong baseline* for advantage estimation—augmented prompts that yield lower rewards than $p_0$ are effectively pruned during policy optimization, preventing wasteful exploration in low-reward regions of the prompt space. Second, consistent exposure to original prompts ensures that the FM maintains robust performance on the training distribution, preventing the model from becoming overly dependent on LM refinements at inference time. We empirically validate the choice of $m$ via ablation in Table 5 and quantify the statistical role of retention across training in Section B.4.

### 4.2. Joint RL Training on Disjoint LMs and FMs

Having established the LM as a dynamic prompt refiner, we now describe the joint RL training procedure that simultaneously optimizes both $\pi_{\text{LM}}$ and $\pi_{\text{FM}}$ within a unified policy gradient framework. Crucially, while the two models share reward signals, they remain architecturally *disjoint*—gradients do not propagate between the LM and FM, preserving modularity and computational efficiency.

At each training iteration, we sample a batch of $B$ original prompts $\{p_0^{(1)}, \ldots, p_0^{(B)}\}$ from the training distribution. For each prompt $p_0^{(j)}$, we generate $n$ samples using the procedure described in Section 4.1: $m$ samples are generated directly from $p_0^{(j)}$ with different noise seeds, while the remaining $n-m$ samples use LM-refined prompts $\{p_i^{(j)}\}_{i=1}^{n-m}$ sampled from $\pi_{\text{LM}}(\cdot|p_0^{(j)})$. Each prompt (original or refined) is paired with an independent latent noise vector $\epsilon \sim \mathcal{N}(0, I)$ to produce images via the FM.

The resulting images are evaluated using a composite reward function $R(\cdot)$ that aggregates format reward and image generation reward:

$$R(\mathbf{x}, p) = \lambda_{\text{Format}} R_{\text{Format}}(p) + \lambda_{\text{Gen}} R_{\text{Gen}}(\mathbf{x}, p), \quad (1)$$

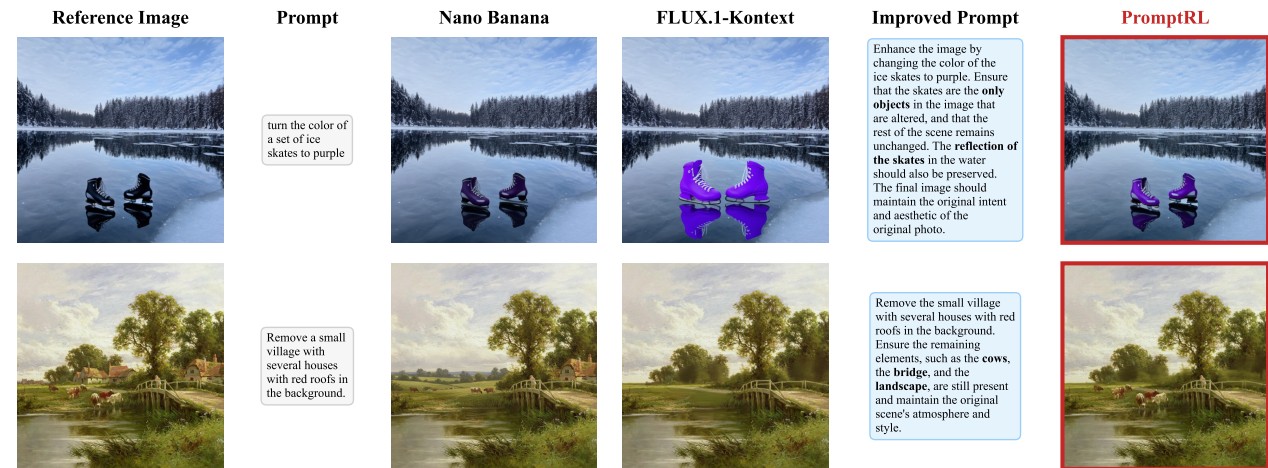

*Figure 3.* Qualitative comparison on instructional image editing tasks. Our method enables the LM to leverage the original image's visual signals to transform vague editing instructions into more explicit and image-specific prompts, ultimately improving editing performance.

where $R_{\text{Format}}(p)$ is a binary reward that requires the LM to output refined prompt variants enclosed within XML tags `<answer>` and `</answer>`. This format constraint ensures structured output parsing and penalizes malformed generations, with $R_{\text{Format}}(p) = 1$ if the output conforms to the required format and 0 otherwise. The term $R_{\text{Gen}}(\mathbf{x}, p)$ denotes an image generation reward model (*e.g.*, GenEval for compositional accuracy). The hyperparameters $\lambda_{\text{Format}}$ and $\lambda_{\text{Gen}}$ balance format compliance and generation quality; in practice, we set $\lambda_{\text{Format}} = 1.0$ and $\lambda_{\text{Gen}} = 1.0$ for simplicity. Following GRPO, we compute advantages through group-wise normalization. For each prompt $p_0^{(j)}$ and its $n$ associated samples $\{\mathbf{x}_1^{(j)}, \ldots, \mathbf{x}_n^{(j)}\}$, we calculate:

$$A(\mathbf{x}_i^{(j)}, p_i^{(j)}) = \frac{R(\mathbf{x}_i^{(j)}, p_i^{(j)}) - \mu^j}{\sigma^j + \epsilon}, \qquad (2)$$

where $\mu^j$ and $\sigma^j$ are the mean and standard deviation of rewards within the $j$-th group, and $\epsilon$ is a small constant for numerical stability. This group-wise normalization makes advantage signals invariant to reward scale across different prompts, enabling stable optimization across diverse semantic categories. It also inherently implements a form of self-competition where samples generated from the same prompt compete against each other, encouraging within-group diversity.

For the language model, we only retain advantages corresponding to the $n - m$ LM-refined samples, excluding those generated from original prompts. The LM policy gradient is:

$$\nabla_{\theta_{\text{LM}}} \mathcal{J}_{\text{LM}} = \mathbb{E}_{p_0} \left[ \sum_{i=m+1}^{n} A(\mathbf{x}_i, p_i) \cdot \nabla_{\theta_{\text{LM}}} \log \pi_{\text{LM}}(p_i | p_0) \right]. \qquad (3)$$

This selective advantage assignment ensures the LM learns

to generate prompt variants that improve upon the baseline. Variants that underperform receive negative advantages and are down-weighted, preventing the LM from introducing counterproductive linguistic changes.

For the flow matching model, we utilize advantages from all $n$ samples (both original and refined prompts) to update the image generation policy:

$$\nabla_{\theta_{\text{FM}}} \mathcal{J}_{\text{FM}} = \mathbb{E}_{p_0} \left[ \sum_{i=1}^{n} A(\mathbf{x}_i, p_i) \cdot \nabla_{\theta_{\text{FM}}} \log \pi_{\text{FM}}(\mathbf{x}_i | p_i, \epsilon_i) \right]. \qquad (4)$$

The inclusion of advantages from original prompts ensures the FM maintains strong performance on the base distribution while benefiting from the expanded exploration space provided by LM refinements.

Notably, this joint optimization requires no architectural/algorithmic modifications to either model—we simply backpropagate separate policy gradients through their respective parameters using shared rewards as the update signal. Our framework is agnostic to the specific RL algorithm; while we use GRPO (Shao et al., 2024) as the baseline implementation for its simplicity, the components of PromptRL should be able to be integrated with other online RL approaches for LMs (*e.g.*, ReMax (Li et al., 2023)) and FMs (*e.g.*, DiffusionNFT (Zheng et al., 2026)).

### 4.3. Multi-Reward Training via Reward Tagging

Beyond single-reward scenarios, we validate PromptRL under multi-reward optimization with GenEval, PickScore, and OCR. A known challenge is that different reward models exhibit vastly different scales and variances, requiring cumbersome manual tuning of reward coefficients. Alternative approaches adopt multi-stage training pipelines (*e.g.*,

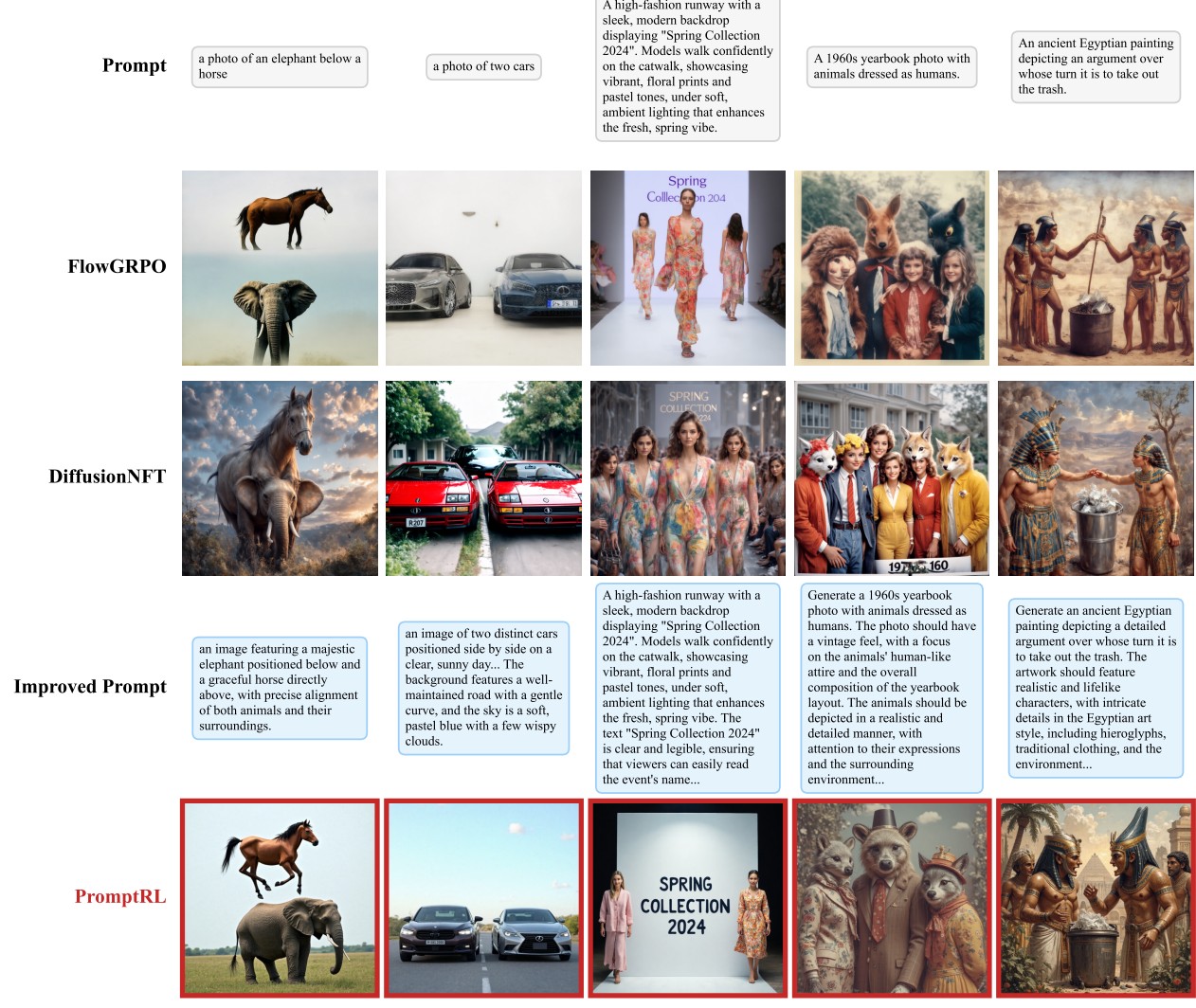

*Figure 4.* Qualitative comparison on text-to-image generation. The first two prompts are from GenEval. The third prompt is from OCR-1k. The last two prompts are from DrawBench.

DiffusionNFT first trains on PickScore before transitioning to additional rewards), introducing scheduling complexity. We propose a simple yet effective solution: *single-reward-per-sample training*. Rather than computing weighted reward sums, we assign each training prompt a categorical tag indicating which reward function evaluates its generated images. During group-wise advantage computation, normalization is performed within each reward category, ensuring comparable advantage signals despite differing scales. This eliminates reward coefficient tuning entirely—each reward model operates in its native scale without interference. Empirically, this simple tagging mechanism achieves strong performance across all objectives simultaneously without explicit reward engineering or multi-stage curricula.

## 5. Experiments

### 5.1. Experimental Setup

**Base models.** To comprehensively validate PromptRL on flow-based image generation, we evaluate on both T2I synthesis and instructional image editing. We adopt FLUX.1-dev as the base flow-matching model for T2I generation and FLUX.1-Kontext for image editing. For the language model component, we use Qwen2.5-VL-3B-Instruct as the prompt refiner.

**Datasets and benchmarks.** For text-to-image generation, following FlowGRPO, we evaluate across three complementary dimensions: compositional accuracy (GenEval), text rendering capability (OCR), and human preference alignment (PickScore). For GenEval and OCR objectives, we

*Table 2.* Performance comparison on GenEval benchmark across different models. Higher scores indicate better performance. Best results are shown in **bold**. Metrics of models with * are obtained from the Qwen-Image paper.

| Model | 1 Obj. | 2 Obj. | Cnt. | Clr. | Pos. | Attr. | Avg.↑ |
|---|---|---|---|---|---|---|---|
| Show-o* (Xie et al., 2025) | 0.95 | 0.52 | 0.49 | 0.82 | 0.11 | 0.28 | 0.53 |
| Emu3-Gen* (Wang et al., 2024b) | 0.98 | 0.71 | 0.34 | 0.81 | 0.17 | 0.21 | 0.54 |
| SD3 Medium* (Esser et al., 2024) | 0.98 | 0.74 | 0.63 | 0.67 | 0.34 | 0.36 | 0.62 |
| FLUX.1-dev* (Labs, 2024) | 0.98 | 0.81 | 0.74 | 0.79 | 0.22 | 0.45 | 0.66 |
| SD3.5 Large* | 0.98 | 0.89 | 0.73 | 0.83 | 0.34 | 0.47 | 0.71 |
| JanusFlow* (Ma et al., 2025) | 0.97 | 0.59 | 0.45 | 0.83 | 0.53 | 0.42 | 0.63 |
| Janus-Pro-7B* (Chen et al., 2025) | 0.99 | 0.89 | 0.59 | 0.90 | 0.79 | 0.66 | 0.80 |
| HiDream (Cai et al., 2025b) | 1.00 | 0.98 | 0.79 | 0.91 | 0.60 | 0.72 | 0.83 |
| Seedream 3.0* (Gao et al., 2025) | 0.99 | 0.96 | 0.91 | 0.93 | 0.47 | 0.80 | 0.84 |
| Qwen-Image* (Wu et al., 2025) | 0.99 | 0.92 | 0.89 | 0.88 | 0.76 | 0.77 | 0.87 |
| *RL-based* | | | | | | | |
| RePrompt (Wu et al., 2026b) | 0.98 | 0.87 | 0.77 | 0.85 | 0.62 | 0.49 | 0.76 |
| FlowGRPO (Liu et al., 2025a) | 1.00 | 0.99 | 0.91 | 0.89 | 0.95 | 0.80 | 0.92 |
| DiffusionNFT (Zheng et al., 2026) | 1.00 | 0.98 | 0.74 | 0.92 | 0.85 | 0.80 | 0.88 |
| PromptRL w/o PE | 1.00 | 0.96 | 0.95 | 0.95 | 0.93 | 0.85 | 0.94 |
| PromptRL w/ PE | **1.00** | **0.99** | **0.99** | **0.96** | **0.99** | **0.90** | **0.97** |

*Table 3.* Performance comparison across different models on aesthetic and OCR metrics.

| Model | Aesthetic | | | OCR | | |
|---|---|---|---|---|---|---|
| | P.S. | HPS | U.R. | OCR-1k | TMDB | OpenLib |
| SD1.5 (Rombach et al., 2022) | 20.92 | 23.71 | 2.00 | 0.05 | 0.13 | 0.08 |
| SDXL (Podell et al., 2023) | 22.14 | 26.67 | 2.78 | 0.13 | 0.20 | 0.09 |
| SD3 Medium (Esser et al., 2024) | 22.38 | 28.56 | 3.09 | — | 0.44 | 0.33 |
| FLUX.1-schnell (Labs, 2024) | 22.64 | 29.39 | 3.25 | 0.54 | 0.66 | 0.50 |
| FLUX.2-klein (Labs, 2025) | 22.79 | 29.03 | 3.29 | 0.55 | 0.22 | 0.46 |
| Z-Image (Cai et al., 2025a) | 20.14 | 28.22 | 3.51 | 0.70 | 0.71 | 0.83 |
| Qwen-Image (Wu et al., 2025) | 23.05 | 30.40 | 3.53 | 0.65 | 0.79 | 0.94 |
| Qwen-Image-2512 | 23.16 | 30.79 | 3.40 | 0.72 | 0.81 | 0.87 |
| *RL-based* | | | | | | |
| FlowGRPO | 23.33 | 29.80 | 3.33 | 0.89 | 0.83 | 0.73 |
| DiffusionNFT | 23.63 | 31.79 | 3.39 | 0.89 | 0.91 | 0.86 |
| PromptRL w/o PE | 24.01 | 31.79 | 3.38 | 0.97 | 0.92 | 0.95 |
| PromptRL w/ PE | 24.05 | 32.03 | 3.44 | 0.98 | 0.91 | 0.95 |

use the corresponding training sets from FlowGRPO. For PickScore optimization, we train on the Pick-a-Pic dataset. For instructional image editing, we randomly sample 10,000 examples from the OmniEdit (Wei et al., 2025) training set, retaining only the editing instructions and corresponding reference images. For T2I generation, we use the official GenEval validation set for compositional evaluation, Draw-Bench (Saharia et al., 2022) for aesthetic quality assessment (PickScore, HPS (Wu et al., 2023), UnifiedReward (Wang et al., 2025d)), and three OCR benchmarks: the FlowGRPO OCR-1k validation set, TMDB, and OpenLib from MARIO-Eval (Chen et al., 2023). For image editing, we directly evaluate on the OmniEdit validation set, which covers six editing categories: Object Swap, Object Addition, Object Removal, Attribute Modification, Environment Change, and Style Transfer.

**Training details.** For computational efficiency, we conduct rollouts at 512×512 resolution with 20 inference steps when training FLUX.1-dev. However, we observe that reducing resolution significantly degrades the editing capabilities of FLUX.1-Kontext; therefore, we maintain 1024×1024 resolution for editing experiments. To improve training efficiency under the high computational demands of high-resolution training, we reduce the number of inference steps to 8 and apply SDE solving only during the first 4 denoising steps of each rollout, following prior work (Li et al., 2025) demonstrating that image structure largely stabilizes in the early timesteps. All models are trained using the GRPO algorithm with a group size of $n = 8$ samples per prompt and a prompt retention number of $m = 2$.

### 5.2. Quantitative Comparison

**Text-to-image generation.** As shown in Table 2, PromptRL achieves state-of-the-art performance on GenEval with an overall score of **0.97**, outperforming FlowGRPO at 0.92 and DiffusionNFT at 0.88. PromptRL w/ PE attains near-perfect

scores of 0.99 on both Position and Counting, demonstrating exceptional compositional accuracy. Notably, even without prompt enhancement at inference, PromptRL w/o PE achieves 0.94, confirming that joint training instills robust visual understanding independent of LM refinements. Table 3 further validates PromptRL across aesthetic and OCR metrics. Our method achieves a PickScore of 24.05, HPS of 32.03, and OCR accuracy of 0.98 on OCR-1k, consistently surpassing prior RL-based approaches. These improvements across diverse metrics demonstrate that joint LM-FM optimization generalizes beyond single-objective reward optimization.

**Instructional image editing.** For instructional image editing, as reported in Table 4, PromptRL w/ PE improves upon the FLUX.1-Kontext baseline from 1.19 to **1.43**, approaching ReasonEdit-Think at 1.44, which relied on fine-grained and dense data annotations along with a complex multi-stage training pipeline. Improvements are most pronounced on Removal (+0.69) and Environment (+0.28). Notably, naively applying prompt enhancement without joint training degrades FLUX.1-Kontext performance from 1.19 to 1.01, demonstrating that PromptRL's joint optimization is essential for effective prompt refinement.

**Multi-reward training.** We evaluate our tag-based multi-reward training strategy in Table 15 (appendix). Despite using no reward coefficient tuning or multi-stage curriculum, the multi-reward model achieves competitive performance across all objectives (GenEval: 0.93, OCR: 0.96, PickScore: 23.94), with only modest degradation compared to single-reward specialists.

### 5.3. Qualitative Comparison

Figs. 3 and 4 presents qualitative comparisons across T2I generation and image editing tasks. For compositional prompts, PromptRL correctly renders objects with accurate colors and spatial arrangements where baselines exhibit color leakage or omission. On OCR tasks, PromptRL produces legible, correctly spelled text while maintaining aes-

*Table 4.* Performance comparison on image editing tasks measured by EditReward. We evaluate models across six editing categories: Swap, Style, Addition (Add.), Attribute Modification (attr.), Environment (Env.), and Removal. The editing instructions of FLUX.1-Kontext w/ PE are refined by pretrained Qwen2.5-VL.

| Model | Swap | Style | Add. | Attr. | Env. | Removal | Avg.↑ |
|---|---|---|---|---|---|---|---|
| InstructPix2Pix (Brooks et al., 2023) | -0.24 | 0.91 | -0.45 | 0.45 | 0.48 | -0.80 | 0.02 |
| MagicBrush (Zhang et al., 2023) | -0.38 | 0.36 | -0.78 | -0.80 | 0.91 | -0.85 | -0.27 |
| LEDITS++ (Brack et al., 2024) | -0.81 | -0.32 | -0.30 | -0.60 | -0.37 | -0.97 | -0.60 |
| Qwen-Image-Edit | 1.11 | 1.14 | 0.95 | 0.90 | 1.39 | 0.61 | 1.03 |
| FLUX.2-klein | 1.42 | 1.73 | 1.29 | 1.42 | 1.80 | 0.32 | 1.34 |
| Nano Banana | 1.58 | 1.20 | 1.28 | 1.18 | 1.61 | 1.13 | 1.37 |
| Step1X-Edit (Liu et al., 2025b) | 1.39 | 1.58 | 1.19 | 1.34 | 1.57 | 0.22 | 1.24 |
| ReasonEdit (Yin et al., 2025) | 1.51 | 1.43 | 1.19 | 1.47 | 1.58 | 1.14 | 1.40 |
| ReasonEdit-Think | **1.52** | **1.47** | 1.19 | **1.44** | 1.69 | **1.27** | **1.44** |
| FLUX.1-Kontext | 1.35 | 1.36 | 1.16 | 1.15 | 1.44 | 0.55 | 1.19 |
| FLUX.1-Kontext w/ PE | 1.35 | 0.97 | 1.04 | 0.48 | 1.22 | 0.65 | 1.01 |
| **PromptRL w/o PE** | 1.45 | 1.46 | 1.28 | 1.35 | 1.56 | 0.98 | 1.36 |
| **PromptRL w/ PE** | **1.47** | 1.43 | **1.29** | 1.39 | **1.72** | 1.24 | **1.43** |

thetic quality. For instructional editing, PromptRL's jointly trained LM produces image-aware prompt refinements by leveraging visual signals from reference images, transforming vague instructions into precise, image-specific prompts that preserve foreground subjects while modifying only intended regions.

## 5.4. Ablation Study

**Joint RL boosts training efficiency.** We compare training dynamics as rollout counts increase, with both methods using FlowGRPO for FM optimization and single-update-per-sample for stability. As shown in Fig. 5a and Fig. 5b, PromptRL consistently achieves higher rewards with fewer

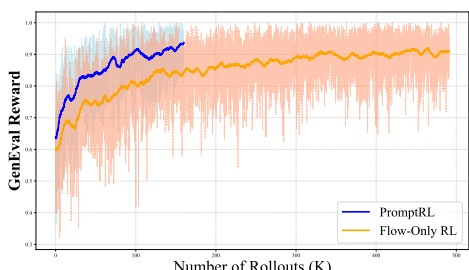

*(a)* Training curve comparison on GenEval reward.

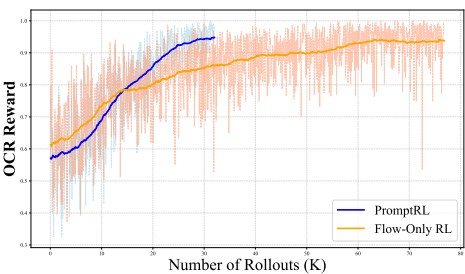

*(b)* Training curve comparison on OCR reward.

*Figure 5.* Training curve comparison on different rewards. Both methods start from the same FLUX.1-dev checkpoint (GenEval 0.66; see Table 2).

*Table 5.* Ablation on prompt retention mechanism. We report GenEval scores without PE at inference with varying numbers of retained original prompts $m$ (group size $n = 8$).

| Retention | 1 Obj. | 2 Obj. | Cnt. | Clr. | Pos. | Attr. | Avg.↑ |
|---|---|---|---|---|---|---|---|
| $m = 0$ | 0.99 | 0.94 | 0.84 | 0.89 | 0.51 | 0.83 | 0.83 |
| $m = 1$ | 0.98 | 0.88 | 0.83 | 0.85 | 0.37 | 0.64 | 0.77 |
| $m = 2$ | **1.00** | **0.96** | **0.95** | **0.95** | **0.93** | **0.85** | **0.94** |
| $m = 4$ | 1.00 | 0.96 | 0.94 | 0.88 | 0.93 | 0.81 | 0.92 |

rollouts across both GenEval and OCR objectives. On GenEval, PromptRL reaches comparable performance to FlowGRPO's convergence point using approximately 50% fewer rollouts. Similar trends are observed on OCR, where PromptRL demonstrates faster initial improvement and superior final performance. These results validate that the expanded exploration space provided by LM-generated prompt variations improves sample efficiency.

**Prompt retention mechanism.** We investigate the effect of retaining original prompts during training on model performance without PE at inference. With group size $n = 8$, we vary the number of retained original prompts $m \in \{0, 1, 2, 4\}$ and evaluate GenEval performance without PE (Table 5). When $m = 0$, the model never sees original prompts during training, resulting in degraded performance on unmodified test prompts (0.83). $m = 1$ yields even worse results (0.77). We empirically find that the LM quickly discovers prompt variants more suitable for the FM, causing the single original prompt to consistently receive negative advantages and thus ineffective gradient updates. Setting $m = 2$ significantly improves performance to 0.94, as multiple original prompts compete within the group, enabling meaningful advantage signals and robust learning on the original prompt distribution. A complementary statistical analysis tracking how often original prompts exceed the group-mean reward throughout training is provided in Section B.4, confirming that retention continues to inject informative gradient signal on the base prompt distribution at every training stage.

## 5.5. Deeper Analyses

We summarize the key analyses motivated by reviewer feedback; per-variant failure modes, the statistical role of retention, and qualitative rewrite examples are deferred to Section B.

**Why joint training?** Table 6 compares PromptRL against decoupled alternatives. A jointly trained 3B LM *without* test-time PE (0.94) already surpasses Flow RL paired with a frozen frontier-LM rewriter (Claude Sonnet 4.6) and test-time PE (0.93), indicating that the benefit comes primarily from LM-FM co-adaptation rather than rewrite quality. Applying any frozen rewriter during RL and evaluating without it actually hurts the FM (0.76–0.78 vs. 0.93 for flow-only

*Table 6.* Training-strategy variants on GenEval. "w/o PE" / "w/ PE" indicate whether the LM rewriter is applied at evaluation; "—" marks inapplicable combinations. Only PromptRL trains the LM and FM in a single RL loop with shared rewards.

| Variant | w/o PE | w/ PE |
|---|---|---|
| Flow-only RL, $2\times$ rollouts | 0.93 | — |
| FLUX.1-dev + frozen Sonnet 4.6 (no RL) | — | 0.74 |
| Flow RL + frozen Qwen2.5-VL-3B rewriter | 0.78 | — |
| Flow RL + frozen Sonnet 4.6 rewriter | 0.76 | 0.93 |
| Sequential: Flow-only RL, then PE-only RL | — | 0.93 |
| **PromptRL** (joint LM–FM RL) | **0.94** | **0.97** |

RL), since the FM becomes dependent on a fixed rewrite distribution absent at test time. Fig. 6 further shows that a sequential pipeline—flow-only RL first, then PE-only RL on the frozen FM—collapses upon introduction of the PE rewriter and recovers only to the 0.93 ceiling, confirming that co-adaptation cannot be replicated post hoc. Under single-reward training, PromptRL also shows no extra reward hacking on held-out rewards (GenEval→OCR: 0.68 vs. 0.69; OCR→GenEval: 0.60 vs. 0.60; see Section B.3).

**Generation diversity is preserved.** Table 7 reports interimage similarity (II-Sim, CLIP ViT-g-14; lower is more diverse) on GenEval prompts under fixed noise. Flow-only RL collapses diversity (0.8298→0.8633), while PromptRL recovers nearly the pre-RL level (0.8336) *while* reaching 0.97 GenEval, showing that linguistic diversification offsets the diversity collapse induced by reward optimization. The KL term ($10^{-2}$) on the LM prevents the rewriter itself from collapsing.

**Wall-clock efficiency.** The 3B LM adds a constant $\sim$6.8 s per step (5.7 s inference + 1.1 s backward) independent of FM denoising depth, amounting to 2.7–7.8% of total step time across 20/10/5-step configurations on $8\times$H100 (Section B.2). Combined with approximately 50% fewer

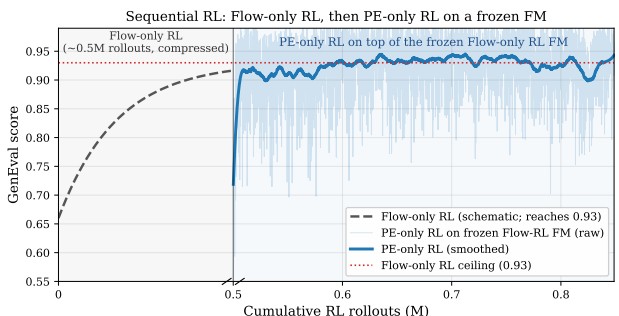

*Figure 6.* Sequential RL: flow-only RL first, then PE-only RL on the frozen FM. Introducing the PE rewriter immediately collapses the reward; PE-only RL spends $\sim$0.1 M rollouts recovering to the 0.93 ceiling and the remaining $\sim$0.24 M rollouts oscillating around it. PromptRL's joint training reaches 0.97 with only 0.25 M rollouts.

*Table 7.* Inter-image similarity (II-Sim, lower is more diverse) on GenEval prompts under fixed noise.

| Setting | II-Sim↓ |
|---|---|
| FLUX.1-dev (no RL) | 0.8298 |
| FLUX.1-dev + frozen LM rewrite | 0.7195 |
| FLUX.1-dev + flow-only RL | 0.8633 |
| FLUX.1-dev + PromptRL (ours) | **0.8336** |

*Table 8.* PromptRL vs. flow-only RL at $2\times$ rollouts.

| Method | Rollouts | GenEval↑ | OCR↑ | PickScore↑ |
|---|---|---|---|---|
| Flow-only RL | $2\times$ | 0.93 | 0.93 | 23.85 |
| **PromptRL** | $1\times$ | **0.97** | **0.98** | **24.05** |

*Table 9.* Generalization of PromptRL's PE to unseen flow models on GenEval. Both PE modules use Qwen2.5-VL-Instruct-3B.

| Model | Original | Ours' PE | Prompt-only RL |
|---|---|---|---|
| SANA | 0.62 | 0.70 | 0.76 |
| SD3 | 0.58 | 0.77 | 0.83 |

rollouts (Table 8), PromptRL is strictly more wall-clock-efficient than flow-only RL.

**Generalization to unseen FMs and compute parity.** The PromptRL-trained PE module transfers to unseen flow models (Table 9), lifting SANA from 0.62 to 0.70 and SD3 from 0.58 to 0.77 on GenEval (1024 resolution, 20 steps). Scaling flow-only RL to twice PromptRL's rollouts still falls short across all metrics (Table 8), confirming that joint optimization reshapes the optimization landscape rather than acting as a compute shortcut. The PE module's generalization further refutes the "patch" framing: a model-specific patch would not transfer between distinct flow backbones.

## 6. Conclusion

We presented PromptRL, a framework that jointly trains language models and flow-matching models within a unified reinforcement learning loop for text-to-image generation. Our approach leverages LM-generated prompt variations to expand the exploration space while simultaneously co-evolving a prompt enhancement module that improves generation quality at inference time. Extensive experiments demonstrate that PromptRL achieves state-of-the-art performance across multiple benchmarks. We discuss limitations of the current framework in Section B.7.

## Impact Statement

This paper presents work whose goal is to advance the field of joint RL training on both language models and diffusion models. There are many potential societal consequences of our work, none of which we feel must be specifically

highlighted here.

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

---

**Algorithm 1** PromptRL: Joint RL Training of LM and FM

---

**Require:** Training prompts $\mathcal{D}$, LM $\pi_{\text{LM}}$, FM $\pi_{\text{FM}}$, reward function $R$, group size $n$, retention number $m$
**Ensure:** Optimized $\pi_{\text{LM}}$ and $\pi_{\text{FM}}$

1: **for** each iteration **do**
2:     Sample batch of prompts $\{p_0^{(j)}\}_{j=1}^B$ from $\mathcal{D}$
3:     **for** each prompt $p_0^{(j)}$ **do**
4:         *// Generate prompt variants*
5:         $\mathcal{P}^{(j)} \leftarrow \{p_0^{(j)}\}^m$    *// Retain $m$ original prompts*
6:         **for** $i = m + 1$ to $n$ **do**
7:           $p_i^{(j)} \sim \pi_{\text{LM}}(\cdot \mid p_0^{(j)})$   *// LM refinement*
8:           $\mathcal{P}^{(j)} \leftarrow \mathcal{P}^{(j)} \cup \{p_i^{(j)}\}$
9:         **end for**
10:       *// Generate images and compute rewards*
11:       **for** each $p_i \in \mathcal{P}^{(j)}$ **do**
12:         Sample $\boldsymbol{\epsilon}_i \sim \mathcal{N}(0, I)$
13:         $\mathbf{x}_i \leftarrow \pi_{\text{FM}}(\cdot \mid p_i, \boldsymbol{\epsilon}_i)$
14:         $r_i \leftarrow R(\mathbf{x}_i, p_i)$
15:       **end for**
16:       *// Group-wise advantage normalization*
17:       $\mu^{(j)}, \sigma^{(j)} \leftarrow \text{mean}(\{r_i\}), \text{std}(\{r_i\})$
18:       **for** $i = 1$ to $n$ **do**
19:         $A_i^{(j)} \leftarrow (r_i - \mu^{(j)})/(\sigma^{(j)} + \epsilon)$
20:       **end for**
21:     **end for**
22:     *// Update LM (only on refined prompts)*
23:     $\theta_{\text{LM}} \leftarrow \theta_{\text{LM}} + \alpha_{\text{LM}} \sum_{j,i>m} A_i^{(j)} \nabla \log \pi_{\text{LM}}(p_i^{(j)} \mid p_0^{(j)})$
24:     *// Update FM (on all samples)*
25:     $\theta_{\text{FM}} \leftarrow \theta_{\text{FM}} + \alpha_{\text{FM}} \sum_{j,i} A_i^{(j)} \nabla \log \pi_{\text{FM}}(\mathbf{x}_i^{(j)} \mid p_i^{(j)}, \boldsymbol{\epsilon}_i)$
26: **end for**

---

## A. Training Details

**Pseudo code.** Algorithm 1 presents the complete PromptRL training procedure. The algorithm alternates between generating prompt variants via the LM, producing images through the FM, and updating both models using group-normalized advantages. Key design choices include the prompt retention mechanism (lines 4–9) that maintains $m$ original prompts per group, and the selective gradient updates that train the LM only on refined prompts while the FM learns from all samples.

**Training configurations.** Table 10 summarizes the hyperparameters and computational resources used across all experiments. We adopt consistent settings where possible to ensure fair comparisons, with task-specific adjustments for learning rates and KL coefficients based on preliminary experiments. For image editing, we use higher resolution (1024×1024) and fewer SDE steps to balance quality and efficiency, as discussed in Section 4.

## B. Additional Empirical Analyses

This appendix consolidates supplementary analyses requested during the review process, complementing the summary in Section 5.5. We expand on (*i*) why joint LM-FM training is necessary, with detailed per-variant failure-mode analysis against frozen-LM and sequential (stage-wise) RL pipelines; (*ii*) the wall-clock cost breakdown of the added LM component; (*iii*) absence of reward hacking under cross-reward evaluation; (*iv*) the statistical role of the prompt retention mechanism; and (*v*) the qualitative behavior of the learned prompt rewriter.

*Table 10.* Training configurations and hyperparameters for PromptRL across different reward objectives. All experiments use GRPO as the base RL algorithm. "—" indicates not applicable or same as default.

| Configuration | Text-to-Image | | | Multi-Reward | Image Editing |
|---|---|---|---|---|---|
| | GenEval | OCR | PickScore | | EditReward |
| *Model Setup* | | | | | |
| Base FM | FLUX.1-dev | FLUX.1-dev | FLUX.1-dev | FLUX.1-dev | FLUX.1-Kontext |
| Language Model | | Qwen2.5-VL-3B-Instruct | | | |
| FM Parameters | | 12B | | | 12B |
| LM Parameters | | | 3B | | |
| *Training Details* | | | | | |
| Training Dataset | FlowGRPO-GenEval | FlowGRPO-OCR | Pick-a-Pic | Mixed | OmniEdit (10k) |
| Training Samples | 50,000 | 19,653 | 25,432 | 95,085 | 10,000 |
| Training Resolution | $512 \times 512$ | $512 \times 512$ | $512 \times 512$ | $512 \times 512$ | $1024 \times 1024$ |
| Training Precision | bfloat16 | bfloat16 | bfloat16 | bfloat16 | bfloat16 |
| Inference Steps (Rollout) | 20 | 20 | 20 | 20 | 8 |
| SDE Steps | 20 | 20 | 20 | 10 | 4 |
| Group Size ($n$) | 8 | 8 | 8 | 8 | 8 |
| Prompt Retention ($m$) | 2 | 2 | 2 | 2 | 2 |
| Batch Size | 1 | 1 | 1 | 1 | 1 |
| K epochs | 1 | 1 | 1 | 1 | 1 |
| Learning Rate (FM) | $3\times10^{-7}$ | $3\times10^{-7}$ | $3\times10^{-7}$ | $10^{-7}$ | $2\times10^{-7}$ |
| Learning Rate (LM) | $10^{-6}$ | $10^{-6}$ | $10^{-6}$ | $10^{-6}$ | $4 \times 10^{-7}$ |
| Optimizer | AdamW | AdamW | AdamW | AdamW | AdamW |
| KL Coefficient (FM) | $4\times10^{-3}$ | $4\times10^{-3}$ | $2\times10^{-3}$ | $2\times10^{-3}$ | $10^{-2}$ |
| KL Coefficient (LM) | $10^{-2}$ | $10^{-2}$ | $10^{-2}$ | $10^{-2}$ | $10^{-2}$ |
| *Reward Configuration* | | | | | |
| Reward Model | GenEval | OCR Accuracy | PickScore | Tag-based | EditReward |
| Format Reward ($\lambda_{\text{Format}}$) | 1.0 | 1.0 | 1.0 | 1.0 | 1.0 |
| Generation Reward ($\lambda_{\text{Gen}}$) | 1.0 | 1.0 | 1.0 | 1.0 | 1.0 |
| Reward Normalization | Group-wise | Group-wise | Group-wise | Per-tag | Group-wise |
| *Training Cost* | | | | | |
| Number of GPUs | 8 | 8 | 8 | 8 | 8 |
| GPU Type | H100 | H100 | H100 | H100 | H100 |
| Training Rollouts | 0.25M | 0.05M | 0.13M | 0.5M | 0.06M |

## B.1. Why Joint Training? Comparison Against Alternative Variants

A natural skepticism is whether the gains of joint LM-FM optimization can be matched by simpler decoupled schemes. We consider three concrete alternatives that train the LM and FM in isolation or in sequence, and show that each suffers from a distinct failure mode that joint training avoids. The variants and their final GenEval scores are summarized in Table 6 (main paper); below we expand on each.

**Variant 1: Frozen LM rewriter + Flow RL.** The simplest decoupled scheme uses a frozen LM as a static prompt rewriter during flow-only RL. We test two such baselines: a frozen Qwen2.5-VL-3B-Instruct (the same backbone PromptRL fine-tunes) and a frontier-class commercial LLM, Claude Sonnet 4.6. Two findings stand out. First, a jointly trained 3B LM *without* test-time prompt enhancement (0.94) already surpasses a frozen frontier LM *with* test-time prompt enhancement (0.93), indicating that the benefit comes primarily from LM-FM co-adaptation rather than from raw rewrite quality. Second, applying any frozen rewriter during RL but evaluating *without* it actually hurts the FM (0.76–0.78 vs. 0.93 for flow-only RL): the FM becomes dependent on a fixed rewrite distribution that is then absent at evaluation time. Joint training avoids this asymmetry by ensuring the FM learns to handle both raw and refined prompts. We also observed that the frozen Qwen2.5-VL-3B frequently emits Chinese characters or off-format tokens despite English-only instructions, producing

corrupted conditioning; the jointly trained LM rapidly learns to honor the `<answer>...</answer>` format constraint and the target language (see Section B.5).

**Variant 2: Sequential RL — Flow-only RL first, then PE-only RL on top.** A second decoupled scheme is a stage-wise pipeline: first train the FM with flow-only RL until convergence, then freeze the FM and train a PE-only RL module on top. The hope is that a strong FM can be further improved by a downstream rewriter that exploits prompt-side gains. In practice this fails because of a phenomenon already documented in Section 3: *flow-only RL aggressively overfits to the training prompt distribution.* The model that scores 0.93 on the original GenEval phrasings does so by memorizing those specific phrasings; semantically equivalent paraphrases collapse its score (e.g., "a photo of three dogs" generalizes well, but "a photo of two dogs chasing another dog" does not). Once the FM is frozen in this state, any rewrite the PE module proposes is, almost by construction, off-distribution and incurs a reward penalty. The PE module is left with no climbable gradient.

We empirically validate this with the sequential-RL run in Fig. 6. We take a flow-only RL checkpoint trained for $\sim$0.5 M rollouts (the 2$\times$ flow-only RL baseline in Table 8, sitting at the 0.93 GenEval ceiling), freeze it, and apply $\sim$0.35 M additional rollouts of PE-only RL on top with the same group size, KL coefficient, and reward function as PromptRL. Introducing the PE rewriter immediately collapses the reward (smoothed GenEval drops to $\sim$0.72 within the first few hundred rollouts), because the frozen flow-RL FM cannot render the off-distribution rewrites the LM proposes. PE-only RL then spends $\sim$0.1 M rollouts learning "safe" rewrites the frozen FM can handle, climbing back to $\sim$0.93; for the remaining $\sim$0.24 M rollouts the smoothed score oscillates around 0.93 with no sustained gain over the flow-only RL ceiling. The PE module spends its entire compute budget recovering from the perturbation it itself introduced, and the only direction it can move is back toward the (semantically trivial) identity rewrite — exactly the failure mode predicted by the prompt-overfitting analysis. By comparison, PromptRL's joint loop reaches 0.97 on the same benchmark, demonstrating that the additional gain depends on *co-adapting* the FM and LM rather than on adding either component post hoc.

**Variant 3: Joint training (PromptRL).** By contrast, PromptRL evolves the FM and LM within a single RL loop. The FM is continually exposed to a moving distribution of LM-generated rewrites and therefore learns to respond to a *distribution* of phrasings rather than memorizing any one of them. The LM, in turn, is rewarded for producing rewrites the current FM can actually use. Co-adaptation reaches an equilibrium that neither frozen nor sequential scheme can attain.

**Takeaway.** The common failure mode of all decoupled variants is an information bottleneck at the interface between components: a frozen LM cannot adapt to what the FM needs (Variant 1), and a frozen FM cannot accommodate what the LM would write (Variant 2). Joint training removes both bottlenecks. More broadly, this is one more data point for the lesson that, in compute-abundant settings, multi-stage pipelines tend to underperform end-to-end optimization wherever both components are trainable—each pipeline seam is an avoidable information bottleneck, not a virtue.

## B.2. Wall-Clock Efficiency Breakdown

We profile per-iteration wall-clock time on 8$\times$H100 GPUs at three FM SDE-step configurations:

*Table 11.* Per-step wall-clock cost (seconds) of PromptRL vs. flow-only RL on 8$\times$H100. "LM overhead" is the marginal cost from the 3B LM (inference + policy update), which is constant in the number of FM SDE steps.

| FM SDE steps | Flow-only | PromptRL | LM overhead | LM % of total |
|:---:|:---:|:---:|:---:|:---:|
| 20 | 248.6 | 255.4 | +6.8 | 2.7% |
| 10 | 135.1 | 141.9 | +6.8 | 4.8% |
| 5 | 80.5 | 86.9 | +6.8 | 7.8% |

## B.3. Cross-Reward Evaluation: No Extra Reward Hacking

A concern when adding an LM in the loop is whether joint optimization accelerates reward hacking against the training reward at the expense of held-out rewards. We train PromptRL and a flow-only RL baseline on a single reward and evaluate on a disjoint reward dimension.

The two methods are within noise on held-out rewards. We attribute this to the regularizing effect of LM-induced prompt diversity: the FM must perform well across a distribution of phrasings rather than overfitting to lexical patterns associated

*Table 12.* Cross-reward evaluation. Each row trains with a single reward and is evaluated on a different held-out reward. Scores are virtually identical between flow-only RL and PromptRL, indicating that joint optimization does not exacerbate reward hacking.

| Training reward | Eval reward | Flow-only RL | PromptRL |
|---|---|---|---|
| GenEval | OCR | 0.69 | 0.68 |
| OCR | GenEval | 0.60 | 0.60 |

with high training reward.

## B.4. Statistical Role of Prompt Retention

Section 4.1 motivates retaining $m$ original prompts per group as a baseline for advantage estimation. To verify that this is not vestigial, we measure the probability that at least one original prompt's reward exceeds the group mean, across five training-progress buckets ($n = 8, m = 2$):

*Table 13.* Probability that at least one of the $m = 2$ retained original prompts exceeds the group mean reward, by training progress (group size $n = 8$). The probability converges to the uniform-distribution expectation ($m/n = 0.25$), confirming that the LM's rewrites do not systematically dominate original prompts and that retention continues to provide a meaningful baseline throughout training.

| Training stage (% of steps) | P(orig. > mean) |
|---|---|
| 0–20 | 0.447 |
| 20–40 | 0.287 |
| 40–60 | 0.234 |
| 60–80 | 0.225 |
| 80–100 | 0.257 |

Early in training, original prompts dominate ($P \approx 0.45$) because the LM has not yet learned useful rewrites; by mid-training the LM catches up and the probability converges to $\approx m/n = 0.25$, the value expected under a uniform reward distribution. Crucially, it never drops to zero: rewrites do not systematically dominate originals, and the FM continues to receive informative gradient signal on the base prompt distribution. This statistic explains why retention is necessary for strong PromptRL w/o PE results (0.94 GenEval, Table 5).

## B.5. Qualitative Behavior of the Learned Prompt Rewriter

A central concern is whether the fine-tuned LM collapses into narrow rewrite patterns (an analogue of prompt linguistic hacking on the LM side). We sample rewrites from three training phases (Init, Mid, Late) on two contrasting tasks: spatial layout (GenEval) and text rendering (OCR).

**Spatial layout (GenEval).** Init rewrites are verbose and hallucinate background scenes; training compresses them into concise, layout-faithful outputs.

- **Init:** original "a photo of a cow above a snowboard" → "A cow adorned with snowflakes, standing gracefully above a pristine snowboard. . . set against a snowy mountain backdrop. . . "

- **Mid:** original "a photo of a car left of a scissors" → "a realistic photograph of a car placed to the left and scissors to the right, with all entities aligned precisely"

- **Late:** original "a photo of an apple above a keyboard" → "an elevated apple in the top center with a keyboard at the bottom, both perfectly aligned"

**Text rendering (OCR).** Init outputs malformed tags or editing suggestions; training teaches the LM to preserve the target string verbatim and append clarity instructions.

- **Init:** target "*Earth Observation Unit*" → "`1. Replace 'boldest' with 'prominently displayed' ...`" (treats the prompt as an editing task — wrong format).

- **Mid:** target "*Puppies for Adoption*" → "...reads 'Puppies for Adoption' in clear, readable font ..."

- **Late:** target "*Get Out Now*" → "...bold crimson red: 'Get Out Now' ...color and size chosen to ensure maximum visibility ..."

*Table 14.* Quantitative summary of rewrite behavior across training phases. "Format" is the fraction of outputs with valid `<answer>` tags; "Spatial keywords" is the rate of explicit spatial words ("left", "positioned"); "Scene hallucination" is the rate of unrelated scene tokens ("garden", "sunset") absent from the original; "Clarity keywords" is the rate of text-rendering instructions ("readable", "bold").

| Metric | Init | Mid | Late |
|---|---|---|---|
| *Spatial layout (GenEval)* | | | |
|     Avg. length (words) | 77.8 | 32.7 | 40.6 |
|     Format compliance | 78.1% | 99.9% | 99.9% |
|     Spatial-keyword rate | 79.7% | 88.1% | 92.8% |
|     Scene hallucination | 0.84 | 0.06 | 0.07 |
| *Text rendering (OCR)* | | | |
|     Avg. length (words) | 56.7 | 60.5 | 63.3 |
|     Format compliance | 43.8% | 99.4% | 99.8% |
|     Clarity-keyword rate | 68.8% | 87.8% | 83.6% |

The RL signal shapes the rewriter in task-appropriate, *opposite* ways: for spatial tasks it learns to *subtract* (remove scene hallucinations, sharpen spatial language); for text-rendering tasks it learns to *add* (append clarity instructions while preserving the original target string). In both cases, format compliance converges to ~99% and Late-phase rewrites retain meaningful stylistic variation, indicating that the KL regularizer effectively prevents mode collapse while RL steers the rewriter toward task-relevant improvements.

## B.6. Multi-Reward Training Ablation

Here we report the single-reward vs. tag-based multi-reward comparison referenced in Section 4. The multi-reward model is trained with the per-sample tag mechanism described in the main paper; single-reward refers to specialists trained separately on each objective.

*Table 15.* Comparison of single-reward and multi-reward training. Single-reward trains separately on each objective, while multi-reward uses tag-based joint optimization.

| Training | GenEval↑ | OCR↑ | PickScore↑ |
|---|---|---|---|
| Single-reward | 0.97 | 0.98 | 24.05 |
| Multi-reward | 0.93 | 0.96 | 23.94 |

## B.7. Limitations

Although PromptRL achieves strong performance through joint LM-FM optimization, the FM and LM develop a degree of co-adaptation during training. Specifically, when replacing our co-trained prompt enhancer with a different LM (*e.g.*, Qwen-3) at inference time, we observe a performance drop on GenEval from 0.97 to 0.84. This indicates that the FM becomes partially specialized to the linguistic patterns produced by its training-time LM partner. We emphasize that this co-adaptation is by design rather than a fundamental flaw—PromptRL explicitly aims to jointly optimize both components for deployment as a unified system, and in practical scenarios where the co-trained LM and FM are used together, this tight coupling translates into the state-of-the-art performance we report. Future work could explore techniques such as multi-LM training or regularization strategies to further improve cross-LM generalization when broader compatibility is desired.

