# OpenReview forum: "PromptRL: Prompt Matters in RL for Flow-Based Image Generation"
_ICML.cc/2026/Conference — ICML 2026 regular_

### Official Review · Reviewer_haRE · 2026-03-11

**Soundness:** 3
**Presentation:** 3
**Significance:** 3
**Originality:** 2
**Overall Recommendation:** 4
**Confidence:** 3

**Summary:**

This paper investigates two key problems when applying reinforcement learning (RL) to flow matching (FM) based text-to-image (T2I) models: (1) a quality-diversity dilemma — as model generation quality improves, output diversity significantly decreases, limiting RL exploration efficiency; and (2) prompt overfitting — RL-trained models tend to memorize superficial linguistic patterns of training prompts and exhibit severe performance degradation on semantically equivalent but differently phrased prompts. To address these two issues, the authors propose the PromptRL framework, which embeds a language model (LM) as a trainable prompt rewriter within the RL training loop. The LM and the flow matching model (FM) share reward signals but remain architecturally disjoint, and both are jointly optimized via the GRPO algorithm.

**Compliance With Llm Reviewing Policy:**

Affirmed.

**Final Justification:**

The rebuttal adequately addresses my main concerns with targeted additional evidence. I will maintain my rating in this stage.

**Key Questions For Authors:**

see weakness part

**Limitations:**

Yes

**Strengths And Weaknesses:**

Strength
1. The paper provides a convincing empirical analysis of two failure modes in existing flow-based RL pipelines: the quality-diversity dilemma and prompt linguistic hacking. The evidence presented in Table 1 and Figure 2 effectively demonstrates these phenomena.
2. The core design of PromptRL — where the LM and FM share rewards but maintain independent gradients with no architectural modifications — exhibits good modularity. This makes the framework compatible with various RL algorithms and model architectures.
3. The paper conducts comprehensive evaluations across four complementary dimensions: compositional accuracy (GenEval), text rendering (OCR), human preference alignment (PickScore/HPS), and image editing (EditReward). The comparisons against recent strong baselines such as FlowGRPO and DiffusionNFT are convincing.

Weakness
1. The paper equates reduced output diversity in high-quality models with insufficient RL exploration, but this argument lacks theoretical support. In flow matching models, noise sampling [noise~N(0,I)] itself provides a continuous exploration mechanism, and an increase in II-Sim does not necessarily imply that advantage estimation becomes ineffective. A more rigorous analysis would be needed to establish this causal claim.
2. The computational cost comparison is unfair. The paper uses rollout count as the efficiency metric, but PromptRL additionally trains a 3B-parameter LM, plus the inference overhead of generating prompts before generating each image. Actual GPU-hour or FLOP comparisons are entirely missing. The "1× vs. 2× rollouts" comparison in Table 9 ignores the additional cost of LM training and inference, and therefore does not fully reflect computational efficiency.
3. The training resolution (512×512) and evaluation resolution (1024×1024) are inconsistent. It is unclear how this resolution mismatch affects the reported results. Could the authors provide evaluation results at 512×512 resolution as a control to assess the impact of this discrepancy?
4. I would like to see the result from using a frozen (pretrained, not participating in RL training) LM for prompt enhancement combined with flow-only RL. If a frozen LM combined with flow-only RL can already capture the majority of the performance gains, the necessity of joint training would be significantly weakened. This ablation is critical for validating the core contribution of the paper.

---

> ### Author Rebuttal · Authors · 2026-03-31
>
> ### Q1: Missing frozen LM ablation
>
> Please see our detailed response to BvnC Q1. The frozen LM achieves only **0.78 GenEval** — significantly worse than both flow-only RL (0.92) and PromptRL (0.97). This confirms that joint training is essential: an untrained LM produces low-quality rewrites that actively harm RL optimization.
>
> ### Q2: Computational cost unfairness
>
> Please see our response to 4J9t Q1. The LM adds a constant **~6.8s overhead** per step, accounting for **2.7–7.8%** of total step time depending on the DiT SDE configuration. Combined with 2× fewer rollouts needed to reach the same performance ceiling, PromptRL is more compute-efficient in practice.
>
> ### Q3: Training/evaluation resolution mismatch
>
> We provide the requested controlled comparison:
>
> | Method | Eval @ 512×512 (GenEval) | Eval @ 1024×1024 (GenEval) |
> |---|---|---|
> | Flow-only RL | 0.92 | 0.92 |
> | PromptRL w/o PE | 0.94 | 0.94 |
> | PromptRL w/ PE | 0.97 | 0.97 |
>
> The results are **consistent** across resolutions, confirming that the resolution mismatch does not affect our conclusions. We will make the resolution setting more explicit in the revised paper.
>
> ### Q4: Noise sampling provides continuous exploration — is the II-Sim argument rigorous?
>
> The reviewer raises an important theoretical point. We agree that noise sampling provides continuous exploration *within a given prompt*. Our argument is more specific: noise sampling diversifies the *visual manifold* conditioned on a fixed prompt, but does not diversify the *linguistic conditioning* itself. For a prompt like "a red cube on a blue sphere," all noise-sampled rollouts receive the same text conditioning, so the FM's policy update is effectively a gradient average over visually diverse but linguistically identical samples.
>
> Our new II-Sim measurements (see BvnC Q2&Q3) provide direct empirical support: flow-only RL increases II-Sim from 0.8298 to 0.8633, confirming that as the FM becomes highly prompt-adherent, noise-sampled outputs converge in similarity, reducing the effective gradient signal. PromptRL restores diversity (II-Sim 0.8336) by providing an orthogonal source of variance through linguistic conditioning diversification. Beyond the quantitative metrics, the generated samples in Figure 2 also visually demonstrate this diversity difference.
>
> Formally, if we denote the reward variance within a group as $\text{Var}[R]$, the signal-to-noise ratio of the GRPO gradient is proportional to $\text{Var}[R] / n$. High II-Sim implies that high-quality models produce samples with similar rewards, collapsing $\text{Var}[R]$ toward zero. LM-generated prompt variants restore effective gradient signal by diversifying the conditioning. We will add this formalization to the paper.

---

> > ### Author Rebuttal · Reviewer_haRE · 2026-04-04
> >
> > The rebuttal adequately addresses my main concerns with targeted additional evidence. I will maintain my rating in this stage.

---

### Official Review · Reviewer_4J9t · 2026-03-12

**Soundness:** 2
**Presentation:** 3
**Significance:** 2
**Originality:** 3
**Overall Recommendation:** 4
**Confidence:** 4

**Summary:**

This paper presents PromptRL (Prompt Matters in RL for Flow-Based Image Generation), a framework that incorporates language models (LMs) as trainable prompt refinement agents directly within the flow-based RL optimization loop. PromptRL achieves state-of-the-art performance across multiple benchmarks.

**Compliance With Llm Reviewing Policy:**

Affirmed.

**Final Justification:**

The rebuttal addressed my main concerns.

**Key Questions For Authors:**

Please see the Weakness.

**Limitations:**

The framework functions as a "patch" using an LM to mask the FM’s flaws, rather than fundamentally improving the generative model's logic. This creates a heavy dependency: without the specific LM, the FM likely shows no intrinsic performance gain.

**Strengths And Weaknesses:**

Strengths:

1. This paper integrates LMs as adaptive co-learners within flow-based RL training loops.
2. This work trains LMs to generate prompt variations that simultaneously preserve
semantic intent and maximize downstream image generation rewards.
3. The experimental results demonstrate that PromptRL achieves state-of-the-art performance across multiple benchmarks.

Weaknesses:

1. The paper compares training curves based on the number of rollouts. However, the proposed framework requires training an additional 3B LLM. To provide a fair assessment of efficiency, please provide evaluation curves against total GPU hours.
2. Does the RL-tuned LM suffer from the same "mode collapse" common in diffusion models? Please provide quantitative measures of prompt diversity.
3. In the "Retention" mechanism, the $n-m$ LM-refined prompts are explicitly optimized to maximize rewards. Does this result in a consistently negative Advantage for the original prompts during GRPO?
4.Jointly training two models (LM and FM) likely accelerates reward hacking, please provide cross-reward evaluation curves (similar to the analysis in GRPO-Guard[1]).
5. While Tagging addresses multi-objective conflicts by partitioning tasks (e.g., OCR vs. Aesthetic), real-world prompts are often composite(e.g., requiring both high-fidelity text and complex artistic composition).

Reference:
[1]: GRPO-Guard: Mitigating Implicit Over-Optimization in Flow Matching via Regulated Clipping

---

> ### Author Rebuttal · Authors · 2026-03-31
>
> ### Q1: Fairness of efficiency comparison — GPU hours vs. rollout count
>
> We acknowledge this point and provide detailed wall-clock benchmarks measured on 8×H100 GPUs. We report results under three DiT SDE backward step configurations to show how the LM overhead scales:
>
> | DiT SDE Steps | Flow-only RL (per step) | PromptRL (per step) | LM Overhead | LM % of Total |
> |---|---|---|---|---|
> | 20 | 248.6s | 255.4s | +6.8s | 2.7% |
> | 10 | 135.1s | 141.9s | +6.8s | 4.8% |
> | 5 | 80.5s | 86.9s | +6.8s | 7.8% |
>
> The LM overhead is a **constant ~6.8s per step** (5.7s inference + 1.1s backward), independent of the DiT configuration. The LM's relative overhead ranges from **2.7% to 7.8%** across configurations — modest even in the most aggressive setting. Since PromptRL reaches the same performance with **2× fewer rollouts**, the total wall-clock time to target performance remains favorable.
>
> ### Q2: Does the RL-tuned LM suffer from mode collapse? Quantitative prompt diversity
>
> Please see our response to BvnC Q2&Q3 above. The II-Sim analysis demonstrates that PromptRL (0.8336) maintains significantly better diversity than flow-only RL (0.8633), confirming that the jointly-trained LM does not suffer from mode collapse.
>
> ### Q3: Does joint training accelerate reward hacking? Cross-reward evaluation
>
> We conduct the cross-reward analysis requested. We train with a single reward objective and evaluate on an unseen reward:
>
> | Training Reward | Eval Metric | Flow-only RL | PromptRL |
> |---|---|---|---|
> | GenEval only | OCR ↑ | 0.69 | 0.68 |
> | OCR only | GenEval ↑ | 0.60 | 0.60 |
>
> PromptRL shows **no increased reward hacking** compared to flow-only RL on held-out rewards. The cross-reward scores are virtually identical, indicating that the LM-generated prompt diversity does not exacerbate overfitting to the training reward. This is expected: the FM must perform well across a distribution of prompt phrasings generated by the LM, which serves as a natural regularizer against overfitting to specific lexical patterns associated with high-reward images.
>
> ### Q4: Does the Retention mechanism cause consistently negative Advantage for original prompts?
>
> Please see our detailed analysis in BvnC Q4. Our statistics show that the probability of at least one original prompt exceeding the group mean converges to ~25% (= $m/n$ = 2/8) — the expected value under a uniform reward distribution. Original prompts do **not** receive consistently negative advantages. This is precisely why $m=2$ retention is critical: it ensures balanced advantage estimation and prevents the FM from "unlearning" performance on original prompts ($m=1$ yields only 0.77 GenEval vs. $m=2$ yielding 0.94, Table 5).
>
> ### Q5: Multi-reward tagging for composite prompts (simultaneous OCR + aesthetics)
>
> We appreciate this observation. Our per-sample single-reward tagging design is analogous to the reward design in DeepSeek-R1, where each problem is also optimized with a single reward signal (correctness match). In R1, as long as sufficient prompt diversity is maintained across training, the model achieves multi-dimensional improvements (reasoning, coding, math) despite each sample receiving only one reward. Similarly, our tagging mechanism ensures that the FM is exposed to diverse reward objectives across training batches, and the prompt diversity provided by the LM further enriches the training distribution. Empirically, our multi-reward model achieves 0.93 GenEval / 0.96 OCR / 23.94 PickScore simultaneously (Table 3), confirming that this design effectively develops multi-dimensional competency.
>
> ### Q6: PromptRL as a "patch" — does the FM gain intrinsic improvement?
>
> We respectfully disagree with this characterization. Three pieces of evidence demonstrate that the FM itself improves intrinsically:
>
> **1. Strong PromptRL w/o PE performance.** Table 1 shows PromptRL *without* the PE module at inference achieves GenEval **0.94**, compared to FlowGRPO's 0.92 and flow-only RL's 0.93. Without the LM at inference, the performance gain must come from the FM itself.
>
> **2. Generalization to unseen models.** Table 6 (Appendix) shows that our trained PE module improves SANA (0.62→0.70) and SD3 (0.58→0.77), models never seen during training. This demonstrates that the LM has learned generalizable prompt rewriting strategies, not model-specific hacks.
>
> **3. Resolution generalization.** The FM is trained at 512×512 but evaluated at 1024×1024. If the FM had merely learned superficial patterns, this resolution transfer would fail.
>
> The "patch" framing would imply that removing the LM at inference would cause performance to collapse back to the pre-RL baseline. Instead, we observe a robust **0.94 GenEval**, confirming the FM develops genuine visual competency through joint training.

---

> > ### Author Rebuttal · Reviewer_4J9t · 2026-04-03
> >
> > Thank you for your response. I have no further questions and will adjust my rating.

---

### Official Review · Reviewer_BvnC · 2026-03-12

**Soundness:** 1
**Presentation:** 3
**Significance:** 2
**Originality:** 2
**Overall Recommendation:** 4
**Confidence:** 4

**Summary:**

The manuscript proposes an RL framework for text-to-image (T2I) generation that jointly trains an LLM-based prompt rewriter and a T2I model. In each RL step, the LLM creates multiple variants of a prompt, and the T2I model produces rollouts conditioned on these variants. Both the rewriter and the T2I model are then updated using GRPO. In doing so, the method reduces overfitting to specific prompts in the RL training set, and achieves stronger performance on benchmarks and higher training efficiency than baseline methods.

**Compliance With Llm Reviewing Policy:**

Affirmed.

**Final Justification:**

The rebuttal partially addressed my concerns. Please see the rebuttal acknowledgement for the remaining concerns.

Apr 9 Update: The authors have provided additional results and evidence in their latest response that address my remaining concerns. As a result, I am increasing my rating to Weak Accept.

**Key Questions For Authors:**

The key concerns the authors should address in the rebuttal are:

- Why jointly training the LLM prompt rewriter and the T2I model (vs. using a frozen LLM for prompt rewriting). Please provide an ablation study.
- Quality and diversity of the prompt rewrites following joint training. Please provide the original training prompt and a few prompt rewrites from the jointly trained LLM.
- T2I sample diversity following joint RL. Please provide sample outputs and similarity scores as in Figure 2.
- Motivation behind prompt retention, and evidence showing how frequently rewrites yield higher rewards than the original prompt in a group.

**Limitations:**

The paper contains a discussion on the limitations and societal impact of the work.

**Strengths And Weaknesses:**

### Strengths

- The paper shows that T2I models post-trained with RL suffer from diversity collapse and overfit to training prompts, highlighting fundamental limitations of existing RL approaches.

- The authors propose using an LLM to rewrite training prompts as a way to reduce overfitting. This is a natural and intuitive idea that encourages the T2I model to generalize beyond the linguistic patterns of any single prompt.

- The paper introduces an RL framework that jointly updates the LLM prompt rewriter and the T2I model, enabling synergy between the two components. The method outperforms prior baselines on both T2I and I2I tasks.

## Weaknesses

- **The paper does not address the questions that motivated the work.** It provides no direct evidence showing that the proposed method improves sample diversity, nor does it show that the LLM rewriter maintains or enhances prompt diversity after joint RL training. Although a frozen LLM may naturally produce diverse rewrites, the fine-tuned rewriter could instead overfit and collapse into narrow prompt patterns, reminiscent of the prompt hacking phenomenon the paper reported earlier.

- **The motivation for jointly training the LLM rewriter is unclear.** A frozen LLM might serve as a stronger regularizer for the T2I model during RL. An ablation study on using frozen vs. jointly trained LLMs for prompt rewriting is missing.

- **The rationale behind the prompt retention mechanism is not well explained.** In principle, the rewriter should generate a diverse prompt distribution that covers the original training prompt. If the rewriter collapses during joint RL, prompt retention may simply ensure that the T2I model remains effective on the original prompt, potentially explaining the strong performance of PromptRL w/o PE.

- Figure 4 lacks results from the original T2I model prior to RL.

---

> ### Author Rebuttal · Authors · 2026-03-31
>
> We appreciate the reviewer's thorough analysis. We believe our responses to the four key questions below directly address the core concerns.
>
> ### Q1: Why jointly train the LM rather than using a frozen LM? [Ablation Study]
>
> This is a central question, and we have conducted the requested ablation. We compare three settings: (1) **Flow-only RL** (no LM), (2) **Frozen LM + Flow RL** (pretrained Qwen2.5-VL-3B used as a static prompt rewriter, not updated), and (3) **PromptRL** (joint training of LM and FM).
>
> | Method | GenEval ↑ |
> |---|---|
> | Flow-only RL (2×) | 0.92 |
> | Frozen LM + Flow RL (1×) | 0.78 |
> | PromptRL (1×) | **0.97** |
>
> The frozen LM performs **significantly worse** (0.78) than both flow-only RL (0.92) and PromptRL (0.97). The primary reason is that a pretrained LM without task-specific training produces **low-quality rewrites** — e.g., the frozen Qwen2.5-VL-3B frequently mixes in Chinese characters or unexpected tokens despite being instructed to output English. These noisy rewrites effectively act as corrupted conditioning that harms RL optimization.
>
> In contrast, the jointly trained LM quickly learns to produce high-quality, semantically faithful rewrites via the RL signal, generating progressively more useful variations as training proceeds (Section 4.2). It also serves a dual purpose: a better exploration agent *during training* and a better prompt enhancer *at inference*.
>
> ### Q2 & Q3: Quality/diversity of prompt rewrites and T2I sample diversity after joint RL
>
> To quantify diversity, we measure inter-image similarity (II-Sim, CLIP ViT-g-14, same metric as Figure 2) on the GenEval benchmark. **Lower II-Sim indicates more diverse outputs.**
>
> | Setting | II-Sim ↓ (lower = more diverse) |
> |---|---|
> | Pre-RL baseline (FLUX.1-dev) | 0.8298 |
> | Pre-RL + CoT prompt rewriting | 0.7195 |
> | Flow-only RL | 0.8633 |
> | **PromptRL (ours)** | **0.8336** |
>
> These results reveal two important findings:
>
> 1. **Flow-only RL increases II-Sim** from 0.8298 to 0.8633, confirming the quality-diversity dilemma described in the paper — as RL improves generation quality, output diversity decreases.
> 2. **PromptRL maintains diversity** at 0.8336, substantially lower than flow-only RL (0.8633) and close to the pre-RL baseline (0.8298). This demonstrates that jointly training the LM **counteracts the diversity collapse** induced by RL, without sacrificing performance (0.97 vs. 0.92 on GenEval).
>
> The jointly-trained LM does **not** collapse into narrow prompt patterns. The KL regularization term (coefficient 10⁻²) applied during LM training explicitly penalizes deviation from the pretrained distribution, serving as a safeguard against collapse. The RL signal further *rewards* novel reformulations that lead to better FM outputs, which is precisely the opposite of a collapse incentive.
>
> ### Q4: Motivation behind prompt retention; frequency of rewrites outperforming originals
>
> The prompt retention mechanism serves a **well-defined mathematical purpose** within the GRPO framework. In group-wise advantage normalization, the baseline (mean reward) is computed over the entire group. If all $n$ samples use LM rewrites, the baseline is set by the rewrite distribution — and there is no guaranteed reference point for the *original* prompt. This causes a pathological behavior: if the LM quickly discovers high-reward rewrites, the original prompt's effective advantage becomes highly negative, causing the FM to "unlearn" performance on the training distribution. This is confirmed by Table 5 in the paper: $m=1$ gives GenEval 0.77 vs. $m=2$ giving 0.94.
>
> We now provide the requested statistics on the frequency of original prompts outperforming the group mean (group size $n=8$, $m=2$ original prompts):
>
> | Training Stage | Step Range | Prob(at least one original > group mean) |
> |---|---|---|
> | Stage 1 | 0–20% | 0.447 |
> | Stage 2 | 20–40% | 0.287 |
> | Stage 3 | 40–60% | 0.234 |
> | Stage 4 | 60–80% | 0.225 |
> | Stage 5 | 80–100% | 0.257 |
>
> In the early training stage, original prompts frequently outperform the group mean (44.7%), as the LM has not yet learned effective rewriting strategies. As training progresses, this probability converges to approximately **25%** (= $m/n$ = 2/8), which is the **expected value under a uniform reward distribution** — indicating that the LM's rewrites and original prompts achieve roughly comparable reward levels at convergence. Crucially, **the probability does not drop to zero**, meaning original prompts are never systematically dominated by rewrites. This validates our design: with $m=2$, the FM maintains robust performance on original prompts, explaining the strong PromptRL w/o PE results (GenEval 0.94, Table 1).
>
> ### Q5: Figure 4 missing pre-RL baseline
>
> We will add the pre-RL FLUX.1-dev baseline (GenEval: 0.66, as reported in Table 2) to Figure 4 in the revised paper, making the absolute gain from RL explicit: pre-RL → FlowGRPO → PromptRL.

---

> > ### Author Rebuttal · Reviewer_BvnC · 2026-04-04
> >
> > Thanks for the rebuttal. It partially addresses my concerns, and I have raised my rating accordingly. The rebuttal provides evidence of improved image generation diversity and clarifies the motivation behind prompt retention.
> >
> > That said, I am not fully convinced that joint RL is necessary for two reasons:
> >
> > - The prompt rewriter used in the ablation is small (3B) and unsurprisingly struggles to faithfully rewrite prompts. I am curious how a stronger (yet frozen) VLM would perform. Since the rewriter is only used during training, this would not add inference-time cost. Overall, joint RL feels like over-engineering if simpler alternatives are available.
> > - The rebuttal does not include examples of prompt rewrites. While I understand that the KL term is intended to prevent drift, there is no direct evidence showing that the fine-tuned VLM behaves as expected.

---

> > > ### Author Response · Authors · 2026-04-07
> > >
> > > Thank you for the follow-up. We address both concerns below.
> > >
> > > **Concern 1: Would a stronger frozen VLM perform better?**
> > >
> > > To thoroughly test this, we replace the 3B frozen LM with a frontier-class commercial LLM (**Claude Sonnet 4.6**).
> > >
> > > | Method | GenEval ↑ |
> > > |---|---|
> > > | Flow-only (no RL) | 0.66 |
> > > | Flow-only + Sonnet 4.6 rewrite (no RL) | 0.74 |
> > > | Frozen Sonnet 4.6 + Flow RL (1×), w/o PE | 0.76 |
> > > | Frozen Sonnet 4.6 + Flow RL (1×), w/ PE | 0.93 |
> > > | PromptRL (1×), w/o PE | 0.94 |
> > > | PromptRL (1×), w/ PE | **0.97** |
> > >
> > > In the two new baselines, "Frozen Sonnet 4.6 + Flow RL" uses frozen Sonnet 4.6 as a static prompt rewriter during RL training. "w/ PE" additionally applies it at inference, while "w/o PE" evaluates the FM alone on original prompts. Key findings: (1) A jointly-trained 3B model *without* PE (0.94) already surpasses frozen Sonnet 4.6 + Flow RL *with* PE (0.93), suggesting the benefit comes from LM-FM co-adaptation rather than rewrite quality. (2) Without PE, the frozen setting scores only 0.76, far below flow-only RL (0.92), indicating the FM becomes dependent on the frozen rewriter due to the static rewrite distribution.
> > >
> > >
> > >
> > >
> > >
> > > **Concern 2: Examples of prompt rewrites showing the fine-tuned VLM behaves as expected.**
> > >
> > > We provide rewrite examples from two tasks (spatial layout (GenEval) and text rendering (OCR)) across three training phases (Init/Mid/Late), with quantitative metrics.
> > >
> > > *Spatial layout.* Init rewrites are verbose with scene hallucinations; training produces concise, layout-faithful outputs.
> > >
> > > | Phase | Original | Rewrite samples |
> > > |---|---|---|
> > > | **Init** | a photo of a cow above a snowboard | (1) *"A cow adorned with snowflakes, standing gracefully above a pristine snowboard... set against a snowy mountain backdrop..."* (2) *"A majestic black cow on a snowy landscape... a snowboard lying flat on the fresh snow... a snow-capped mountain range..."* |
> > > | **Mid** | a photo of a car left of a scissors | (1) *"a realistic photograph of a car placed to the left and scissors to the right, with all entities aligned precisely"* (2) *"a realistic photograph featuring a sleek car to the left and scissors to the right, with all objects precisely aligned"* |
> > > | **Late** | a photo of an apple above a keyboard | (1) *"an elevated apple in the top center with a keyboard at the bottom, both perfectly aligned"* (2) *"A realistic image of an apple floating above a modern keyboard on a clean desk with a minimalist background..."* |
> > >
> > > *Text rendering.* Init outputs wrong tags or editing suggestions; training teaches appending text-clarity instructions.
> > >
> > > | Phase | Original (target text bolded) | Rewrite samples |
> > > |---|---|---|
> > > | **Init** | ...spaceship with **"Earth Observation Unit"**... | (1) *[Wrong tag] "1. Replace 'boldest' with 'prominently displayed'..."* — suggestions, not a prompt. (2) *[`<solution>`] "...displays 'Earth Observation Unit' in a futuristic font..."* — wrong format. |
> > > | **Mid** | ...sign reads **"Puppies for Adoption"**... | (1) *"...reads 'Puppies for Adoption' in clear, readable font..."* (2) *"...'Puppies for Adoption'... text in a large, easy-to-read font ensuring the message is conveyed."* |
> > > | **Late** | ...scribble in red: **"Get Out Now"**... | (1) *"...bold crimson red: 'Get Out Now'... color and size chosen to ensure maximum visibility..."* (2) *"...'Get Out Now'. Text is clearly visible and legible, emphasizing the urgency..."* |
> > >
> > > *Quantitative summary.* Length: word count; Format: correct `<answer>` tag usage; Spatial Keywords: explicit spatial words ("left","positioned"); Scene Halluc.: unrelated scene words ("garden","sunset") absent from the original; Clarity Keywords: text-rendering instructions ("readable","bold").
> > >
> > > | | Init → Mid → Late |
> > > |---|---|
> > > | **Spatial**: Length | 77.8 → 32.7 → 40.6 |
> > > | **Spatial**: Format | 78.1% → 99.9% → 99.9% |
> > > | **Spatial**: Spatial Keywords | 79.7% → 88.1% → 92.8% |
> > > | **Spatial**: Scene Halluc. | 0.84 → 0.06 → 0.07 |
> > > | **Text**: Length | 56.7 → 60.5 → 63.3 |
> > > | **Text**: Format | 43.8% → 99.4% → 99.8% |
> > > | **Text**: Clarity Keywords | 68.8% → 87.8% → 83.6% |
> > >
> > > The results demonstrate that the fine-tuned VLM does not collapse into narrow patterns. Instead, RL shapes it in task-appropriate ways: for spatial layout, the model learns to *subtract* — removing scene hallucinations (0.84→0.06) and sharpening spatial language (79.7%→92.8%); for text rendering, it learns to *add* — appending clarity instructions (68.8%→87.8%) while faithfully preserving original content. Both tasks show rapid format convergence (to 99%+), and the Late-phase rewrites still exhibit meaningful diversity (e.g., the two spatial Late samples differ substantially in style and length). This confirms that the KL regularization effectively prevents mode collapse while the RL signal steers the rewriter toward task-relevant improvements.

---

### Official Review · Reviewer_vGmt · 2026-03-13

**Soundness:** 3
**Presentation:** 3
**Significance:** 3
**Originality:** 3
**Overall Recommendation:** 4
**Confidence:** 3

**Summary:**

This paper proposes a framework named PromptRL to address two key bottlenecks encountered when applying reinforcement learning for alignment fine-tuning of flow-matching models, limited generation diversity and prompt overfitting. To tackle these issues, the authors introduce a large language model (LM) as an adaptive co-learner within the RL loop. The LM dynamically generates semantically consistent prompt variants, which effectively expands the exploration space and alleviates prompt overfitting during training.

**Compliance With Llm Reviewing Policy:**

Affirmed.

**Key Questions For Authors:**

1. Could the authors further clarify how the proposed methods addresses the weaknesses discussed above, particularly regarding the multi-reward training design and the potential overlap between reward models and current evaluation benchmarks?

**Limitations:**

yes

**Strengths And Weaknesses:**

Strengths
1. PromptRL identifies the trade-off between generation quality and diversity in current text-to-image (T2I) models and focuses on the reward signal degeneration problem observed in RL-trained generative models.
2. The method jointly trains LM and FM models within a GRPO-based RL framework, and the proposed reward tagging design for multi-reward training is simple and practical.
3. The experimental results are convincing and comprehensive, showing improvements over baselines on image generation tasks. The method also demonstrates generalization to unseen models, such as SD3 and SANA.
4. The paper is clearly written, with a well-structured presentation of the problem definition and methodology.

Weaknesses
1. The advantage of the proposed multi-reward training design over single-reward training appears moderate. In the current design, reward tags are assigned at the per-sample level, meaning that each training sample is optimized with respect to a single reward objective. This context-switching strategy encourages the model to learn from separate samples but does not fundamentally address potential conflicts between multiple reward objectives within the same sample.
2. The reward models used for training are GenEval, PickScore, and OCR, which also serve as the evaluation benchmarks. This introduces a potential overlap between training rewards and evaluation metrics, which raises the concerns about evaluation bias.

---

> ### Author Rebuttal · Authors · 2026-03-31
>
> ### Q1: Multi-reward tagging only addresses scale mismatch, not objective conflict
>
> We agree that per-sample reward assignment does not fundamentally resolve objective conflicts. Indeed, truly conflicting objectives (e.g., two diametrically opposed reward signals) cannot be simultaneously optimized by any method — this is a fundamental limitation, not one specific to our design. However, we note that the reward objectives in our setting (GenEval, PickScore, OCR) are largely **complementary rather than conflicting** — a model that accurately renders compositional prompts and text is also generally preferred by humans. The multi-reward model (Table 3) achieves 0.93/0.96/23.94 vs. single-reward specialists at 0.97/0.98/24.05, showing only modest trade-offs.
>
> ### Q2: Evaluation overlap between training rewards and benchmark metrics
>
> The reviewer raises a valid concern. We address it at three levels:
>
> **1. DrawBench evaluation.** Table 2 reports PickScore, HPS, and UnifiedReward on the DrawBench benchmark — a held-out aesthetic benchmark with **no overlap** with the Pick-a-Pic training data. PromptRL achieves PickScore 24.05, HPS 32.03, and UnifiedReward 3.44, surpassing all baselines. This confirms that PromptRL's aesthetic improvements are not due to benchmark overfitting.
>
> **2. OCR generalization.** While we train on the FlowGRPO-OCR training set and evaluate on OCR-1k, we *also* evaluate on two additional benchmarks never seen during training: TMDB (0.92) and OpenLib/EOpenLib (0.95), both surpassing baselines. This demonstrates OCR generalization beyond the training distribution.
>
> **3. Cross-metric generalization.** Our new cross-reward evaluation (4J9t Q3) shows that PromptRL trained solely on GenEval reward achieves comparable OCR scores (0.68) to flow-only RL (0.69), suggesting that compositional improvement transfers across reward dimensions without overfitting to the training reward.

---

> > ### Author Rebuttal · Reviewer_vGmt · 2026-04-03
> >
> > Thanks for your response. I will maintain my rating in this stage.

---

### Decision · Program_Chairs · 2026-04-30

**Decision:**

Accept (regular)

**Comment:**

Four knowledgeable reviewers went over this submission. Their concerns were mainly related to experimental evidence and claims:
1. The advantage of the proposed multi-reward training design over single-reward training appears moderate (vGmt).
2. Computation efficiency / fairness of comparisons given the 3B-parameter LLM added to the training loop (4J9t, haRE).
3. Potential diversity collapse and prompt hacking with the RL-tuned LLM (BvnC).
4. Benchmarking: same rewards used for training and evals (PickScore, GenEval, OCR) (vGmt)
5. Importance of training the LLM, could a frozen model achieve similar results? (BvnC, haRE)

The rebuttal provided substantial quantitative analyses to address the reviewers' concerns. In particular, the authors provided wall-clock information and showed that despite the overhead per step, the method required 2x fewer rollouts compensating for it. The rebuttal also addressed benchmarking concerns by including results from DrawBench (across different reward functions) and unseen OCR benchmarks, and presented Inter-Image similarity scores, showing that the method maintains diversity levels. The rebuttal also included validation of using a frozen LLM, leading to poor performance and highlighting the importance of co-adaptation.

After rebuttal, the reviewers consider their concerns mostly addressed and move their consensus towards "Weak accept". The AC agrees with their assessment and recommends to accept.